# Liquid biopsy diagnostics for non-small cell lung cancer via elucidation of tRNA signatures
Zhuokun Feng[1,2,3], Masaki Nasu[1], Gehan Devendra[4], Ayman A. Abdul-Ghani[5], Owen T. M. Chan[6], Jeffrey A. Borgia [7], Zitong Gao[1], Hanqiu Zhang[1], Yu Chen [1], Ting Gong[1], Gang Luo[8], Hua Yang [1], Lang Wu [9] ✉, Yuanyuan Fu[1,3,9] ✉ & Youping Deng [1,3,9] ✉

## Abstract

**Background** Lung cancer, particularly non-small cell lung cancer (NSCLC), accounts for about 85% of all lung cancer cases and remains a major global health challenge. Traditional diagnostic methods, such as chest X-rays and low-dose CT scans, have limitations, including high false-positive rates, radiation risks, and the invasiveness of tissue biopsies. This study aims to develop a non-invasive liquid biopsy approach for early NSCLC diagnosis.

**Methods** We developed a machine-learning model to analyze small RNA sequencing data from 1446 tissue samples to identify a diagnostic tRNA signature. This signature was independently validated using the in-house data of 233 plasma exosome samples. Diagnostic performance was assessed using Area Under the Curve (AUC) metrics. Signature tRNAs were then evaluated across various clinical and demographic variables, with further survival analysis and functional studies to explore the molecular role of the signature tRNAs.

**Results** We identify a robust six-tRNA signature with strong diagnostic performance, achieving AUC values of 0.97 in discovery, 0.96 in hold-out validation, and 0.84 in independent validation. The signature effectively distinguishes cancerous from benign samples (AUC = 0.85) and consistently performs across clinical and demographic variables, with AUC values exceeding 0.80, particularly for early-stage lung cancer diagnosis. Additionally, three signature tRNAs demonstrate prognostic value for independent survival prediction. Functional studies suggest potential regulatory roles of specific tRNAs and their associated fragments in tumor metabolism pathways.

**Conclusions** This research underscores the diagnostic power of tRNA signature for NSCLC liquid biopsy and provides epigenetic insights that enhance our understanding of oncogenic molecular pathophysiology.

## Plain Language Summary

Lung cancer is one of the leading causes of cancer-related deaths worldwide, and early detection is crucial for improving patient outcomes. This study investigates whether transfer RNAs (tRNAs), key regulators of protein synthesis, can be used as a non-invasive way to detect non-small cell lung cancer (NSCLC). We analyzed large-scale small RNA sequencing datasets derived from diverse NSCLC cohorts and identified six tRNAs that could be combined to accurately distinguish people with lung cancer from healthy individuals, particularly people with early-stage lung cancer. Further analysis suggests that some of these tRNAs may also play a role in tumor growth. These findings represent a promising step toward developing an easier method to screen for lung cancer, potentially improving early diagnosis and patient survival.

Lung cancer remains a principal public health challenge worldwide, marked by its substantial prevalence and mortality rates. As reported in the 2024 Cancer Statistics, there are 234,580 new instances of lung cancer and 125,070 resultant fatalities in the United States alone, establishing lung cancer as the leading cause of cancer-related mortality[1]. This malignancy is histologically divided into small cell lung cancer (SCLC) and NSCLC, with the latter accounting for approximately 85% of incidences. NSCLC exhibits notable

heterogeneity, subdivided into lung squamous cell carcinoma (LUSC) and lung adenocarcinoma (LUAD). NSCLC presents a high risk for early metastasis, contributing to a grim five-year survival rate of merely 28% within the U.S. population[2]. These statistics underscore the importance of early diagnosis in improving patient survival and treatment outcomes.

Traditional diagnostic avenues, such as chest radiography and low-dose computed tomography (LDCT), are hindered by considerable

drawbacks, including elevated false-positive rates and potential radiation risks[3]. Moreover, the invasive nature and poor compliance associated with multiple tissue biopsies exacerbate the challenges in early cancer detection[4]. Thus, there is an urgent necessity for the innovation of non-invasive and accurate diagnostic methodologies for early lung cancer detection.

In this context, liquid biopsy emerges as a pioneering approach in cancer screening[5,6]. This methodology facilitates a non-invasive, comprehensive evaluation of an individual's overall health in real time by analyzing tumor biomarkers present in body fluids such as blood. It holds potential for diagnosis, prognosis, and monitoring of treatment responses in cancer patients. Predominantly, circulating tumor cells, DNA, and exosomes have been identified as key biomarkers in liquid biopsies[7].

Exosomes, a subset of extracellular vesicles (EVs) with diameters ranging from 30 to 200 nm, serve as crucial mediators in intercellular communication, transporting DNA, mRNA, small non-coding RNA (sncRNA), long non-coding RNA (lncRNA), and proteins. Their important function in many aspects of carcinogenesis and cancer development processes, including angiogenesis, metastasis, epithelial-mesenchymal transition (EMT), and inhibiting antitumor immunity, makes them a focus of liquid biopsy research for lung cancer[3,7–9]. With advances in next-generation sequencing (NGS) technologies, such as small-RNA sequencing, researchers have identified exosomal sncRNAs as vital mediators of information exchange between cancerous and normal cells in lung cancer[10]. For example, *miR-30e-3p*, *miR-30a-3p*, *miR-181-5p*, and *miR-361-5p* are identified to be the diagnostic biomarkers of LUAD[11]; *miR-15b-5p*, *miR-10b-5p*, *miR-320b* are validated to test LUSC[11]; *piR-hsa-26925* and *piR-hsa-5444* are discovered to be the potential biomarkers of LUAD[12]; exosomal tRNA-derived fragments (tRFs) like *tRF-Leu-TAA-005*, *tRF-Asn-GTT-010*, *tRF-Ala-AGC-036*, *tRF-Lys-CTT-049*, and *tRF-Trp-CCA-057* are significantly downregulated in NSCLC patients that can serve as promising diagnostic biomarkers for NSCLC[13]. As a result, blood exosomal sncRNA has great potential to be developed as the diagnostic signature in detecting NSCLC.

Transfer RNA (tRNA), a type of sncRNA, plays an essential role in protein synthesis by transforming amino acids in the mRNA translation process. While it has been traditionally viewed as the housekeeping gene due to its stable expression, recent studies have illuminated the variability in tRNA expression across different organs and between healthy individuals and cancer patients[14]. Moreover, tRNA and its associated metabolism have been discovered to participate in tumorigenesis and cancer progression, including the processes of tRNA transcription and mutations, tRNAs or its derivatives, tRNA modification, and tRNA aminoacylation[15]. The transcription of tRNA mediated by RNA polymerase III (Pol III), which is initially inhibited by tumor-suppressing genes, is aggravated by the products of oncogenes[16]. Telomerase reverse transcriptase (TERT) can exacerbate tRNA transcription by facilitating the binding of Pol III to tRNA genes, thereby enhancing Pol III's occupancy rate. This process is significantly correlated with the upregulation of several tRNA expressions, such as $tRNA^{Lys}$, $tRNA^{Met}$, and $tRNA^{Arg}$, playing a notable role in cancer progression[17]. Kuang et al. discover $tRNA^{Ile}$, $tRNA^{Pro}$, and $tRNA^{Lys}$ are related to tumor differentiation, and patients with up-regulated expression of $mt$-$tRNA^{Glu}$ and $tRNA^{Tyr}$, or down-regulated expression of $tRNA^{Thr}$ and $tRNA^{Asn}$ are more likely to relapse[18]. Furthermore, they also find that $tRNA^{Lys}$, $mt$-$tRNA^{Ser}$, and $tRNA^{Tyr}$ are accurate prognostic biomarkers to predict the survival span of LUAD patients[18].

Despite the growing evidence in tRNA expression between NSCLC patients and healthy individuals, the development of tRNAs as diagnostic signatures and the exploration of their diagnostic efficacy remain limited. Additionally, the relationship between tRNA biomarkers and blood exosomes in lung cancer has yet to be elucidated, posing a barrier to the clinical application of tRNA signatures in liquid biopsies for early cancer screening.

In this study, we address these gaps by employing a machine learning-based approach to identify a six-tRNA diagnostic signature using small RNA sequencing data from multiple global NSCLC cohorts. We then validate the diagnostic performance of this tRNA signature in independent blood exosome datasets, demonstrating its potential utility in non-invasive NSCLC detection. Robustness assessments across various clinical and demographic subgroups, such as age, sex, smoking history, histological subtype, and cancer stage, further confirm the generalizability of the signature. In addition, survival analyses and functional enrichment investigations reveal the prognostic significance of the identified tRNAs and provide insights into their potential regulatory roles in NSCLC pathogenesis.

## Methods

### Data mining from public resources
In this study, we conducted a comprehensive data mining process across the Gene Expression Omnibus (GEO) and The Cancer Genome Atlas (TCGA) databases to identify small RNA sequencing datasets relevant to NSCLC research. Raw sequencing files in FASTQ format, derived from lung tissue samples, were systematically retrieved from six publicly available study cohorts across both databases. These included four GEO datasets—GSE110907 (Korea, URL: https://www.ncbi.nlm.nih.gov/geo/query/acc.cgi?acc=GSE110907)[19], GSE62182 (Canada, https://www.ncbi.nlm.nih.gov/geo/query/acc.cgi?acc=GSE62182)[20], GSE83527 (Canada, URL: https://www.ncbi.nlm.nih.gov/geo/query/acc.cgi?acc=GSE83527)[21], and GSE175462 (Canada, URL: https://www.ncbi.nlm.nih.gov/geo/query/acc.cgi?acc=GSE175462)[22]—and two TCGA datasets: TCGA-LUAD and TCGA-LUSC (USA, https://portal.gdc.cancer.gov/analysis_page?app = ).

In addition to genomic data, corresponding clinical and demographic information was collected, including age, sex, ethnicity, diagnosis, histological subtype, American Joint Committee on Cancer (AJCC) pathological stage, and smoking history. Following data mining, the filtering step was applied to eliminate duplicate and low-quality samples. This process yielded a refined dataset of 1446 samples, including 1173 tumor and 273 normal samples, as detailed in Data S1.

### Plasma sample collection
Plasma specimens and associated patient information were obtained from RUSH University Medical Center, comprising a cohort of 117 individuals diagnosed with NSCLC, 54 subjects with benign lung conditions, and 62 healthy controls, totaling 233 samples (Table 1). The samples were collected between February 2010 and June 2019, and all blood samples were collected at the time of initial diagnosis, forming a retrospective convenience series. While some patients underwent surgical biopsy during the diagnostic phase, any additional treatments—including surgery, radiation, or chemotherapy—were conducted only after blood sampling. Although data regarding immunotherapy treatments were unavailable, the timing of blood collection at diagnosis suggests minimal impact from anti-tumor treatments on the plasma samples. The clinicians responsible for the clinical diagnosis were blinded to the tRNA signature analysis results during the diagnostic evaluation. Histopathological and clinical diagnoses were independently determined based on established clinical and pathological criteria, ensuring an unbiased reference standard assessment.

The ethical acquisition and utilization of human blood samples, along with the clinical data, received approval from the University of Hawaii Human Studies Program, and informed consent was obtained from all patients, under the protocol number 2018-00636.

### Exosome isolation and RNA extraction
Exosomes were meticulously isolated from each plasma specimen utilizing the Capturem Extracellular Vesicle Isolation Kit (Mini, Catalog No. 635741, Takara Bio USA, Inc.). From an initial volume of 500 μL of plasma, the exosomes were subsequently eluted in 200 μL of elution buffer and visualized under an electron microscope (Fig. S1). The miRNeasy Serum/Plasma Kit (Catalog No. 217184, Qiagen, Germany) was used to extract total RNA from these exosome preparations. The resultant RNA was eluted in 16 μL of RNase-free water and preserved at a temperature of −50 °C for subsequent analytical and application purposes.

**Table 1 | Demographic and clinical characteristics of individuals included in the study**

| | | Tissue | | | | | Exosome | *Total* |
|---|---|---|---|---|---|---|---|---|
| | | GSE110907 | GSE62182 | GSE83527 | GS-E175462 | TCGA | RUSH | |
| **Diagnosis** | Lung cancer | 48 | 35 | 36 | 63 | 991 | 117 | 1290(76.83%) |
| | Normal | 48 | 21 | 36 | 77 | 91 | 62 | 335(19.95%) |
| | Benign | 0 | 0 | 0 | 0 | 0 | 54 | 54(3.22%) |
| | *Total* | 96 | 56 | 72 | 140 | 1082 | 233 | 1679 |
| **Age** | <50 | 14 | NA | NA | NA | 53 | 5 | 72(5.21%) |
| | 50–59 | 34 | NA | NA | NA | 189 | 36 | 259(18.74%) |
| | 60–69 | 28 | NA | NA | NA | 364 | 97 | 489(35.38%) |
| | 70–79 | 20 | NA | NA | NA | 379 | 81 | 480(34.73%) |
| | ≥80 | 0 | NA | NA | NA | 70 | 12 | 82(5.93%) |
| | *Total* | 96 | NA | NA | NA | 1055* | 231* | 1382 |
| **Sex** | Female | 96 | 34 | 30 | 100 | 437 | 124 | 821(48.90%) |
| | Male | 0 | 22 | 42 | 40 | 645 | 109 | 858(51.10%) |
| | *Total* | 96 | 56 | 72 | 140 | 1082 | 233 | 1679 |
| **Race** | White | 0 | NA | NA | NA | 799 | 163 | 962(78.02%) |
| | African American | 0 | NA | NA | NA | 88 | 70 | 158(12.81%) |
| | Asian | 96 | NA | NA | NA | 17 | 0 | 113(9.16%) |
| | *Total* | 96 | NA | NA | NA | 904* | 233 | 1233 |
| **Histological subtype** | LUAD | 48 | 35 | 36 | 63 | 513 | 50 | 745(57.84%) |
| | LUSC | 0 | 0 | 0 | 0 | 478 | 46 | 524(40.68%) |
| | Other types of NSCLC | 0 | 0 | 0 | 0 | 0 | 19 | 19(1.48%) |
| | *Total* | 48 | 35 | 36 | 63 | 991 | 115* | 1288 |
| **AJCC pathologic stage** | Stage I | 30 | 18 | 10 | 41 | 507 | 82 | 688(55.48%) |
| | Stage II | 6 | 10 | 10 | 13 | 279 | 6 | 324(26.13%) |
| | Stage III | 12 | 5 | 5 | 5 | 164 | 2 | 193(15.56%) |
| | Stage IV | 0 | 2 | 1 | 1 | 30 | 1 | 35(2.82%) |
| | *Total* | 48 | 35 | 26* | 60* | 980* | 91* | 1240 |
| **Smoking history** | Yes | 14 | 48 | 66 | 94 | 827 | 117 | 1166(73.61%) |
| | No | 82 | 8 | 6 | 46 | 255 | 21 | 418(26.39%) |
| | *Total* | 96 | 56 | 72 | 140 | 1082 | 138* | 1584 |

Bold font highlights column and row titles. The total number of cases in specific categories is presented in *italic formatting*. "*" indicates missing data in certain datasets, with only available information summarized. *LUAD* lung adenocarcinoma, *LUSC* lung squamous cell carcinoma, *NSCLC* non-small cell lung cancer, *AJCC* American Joint Committee on Cancer.

## Library preparation and small RNA sequencing

Small RNA sequencing of all 233 extracted RNA samples was conducted by the Genomics and Bioinformatics Shared Resources (GBSR) at the University of Hawaii Cancer Center, employing the QIAseq miRNA Library Kit (Catalog No. 331502, Qiagen, Germany) for library preparation. The sequencing process utilized the NextSeq 500 System (Illumina, USA) to achieve a target of 10 million reads per sample. This strategic approach ensured a comprehensive analysis of the small RNA profiles within each specimen.

Data retrieved from public databases, including GEO datasets of GSE110907, GSE62182, GSE83527, GSE175462 and TCGA dataset, were all generated using miRNA-seq based on whole transcriptome sequencing on the Illumina HiSeq 2000 platform (Homo sapiens).

## Small RNA sequencing data processing

With reference to the previous pipeline developed by our lab for small RNA seq upstream analysis[23], we developed the pipeline to quantify tRNA for this study. Quality control measures were implemented on all small RNA sequencing raw FASTQ files utilizing FastQC (version 0.11.9)

for initial assessment and fastp (version 0.20.0) for adaptor trimming. This procedure was applied uniformly to datasets obtained from public repositories as well as to the independent cohort collected from RUSH University Medical Center. Subsequent alignment of the processed reads was conducted against the Genome Reference Consortium Human Build 38 (GRCh38/hg38) using the Spliced Transcripts Alignment to a Reference (STAR) software (version 2.7.10). To perform gene feature annotation and the quantification of mapped reads specific to tRNA and tRF, FeatureCounts (version 2.0.6) was employed in conjunction with customized Gene Transfer Format (GTF) files for each RNA gene. Specifically, the tRNA GTF file was derived and compiled from hg38 tRNA fasta files available at GtRNAdb (http://gtrnadb.ucsc.edu/). In parallel, tRF annotations were transformed from the hg19 tRF metadata provided by MINTbase (https://cm.jefferson.edu/MINTbase/) to the hg38 tRF reference using hgLiftOver (https://genome.ucsc.edu/cgi-bin/hgLiftOver), then summarized into a customized hg38 tRF GTF file. Following annotation, the raw count data for tRNAs were converted into Transcripts Per Million (TPM) format, facilitating downstream analytical processes.

## Principal component analysis (PCA)

Gene counts exhibiting low expression, defined as being absent in over 50% of samples within a specific study cohort, were systematically excluded. This threshold sets a balance between retaining relevant features and eliminating genes with low expression levels that might not have substantial discriminatory value, especially in clinical screening applications. The limma package (version 3.56.2) was employed for data normalization to address batch effects across different studies. A preliminary analysis of tRNA expression across all samples was conducted using Principal Component Analysis (PCA), with the resulting expression patterns visualized in a two-dimensional space defined by the first two principal components via ggplot2 package (version 3.4.4) in R. Given the non-normal distribution of PCA results, a rank transformation was applied prior to performing a Multivariate Analysis of Variance (MANOVA). This analysis aimed to identify significant differences in tRNA expression patterns between tumor samples and control groups, which included both normal and benign specimens. Additionally, comparisons of tRNA expression profiles between different NSCLC subtypes were conducted to explore potential subtype-specific variations.

## Model development to identify and validate the tRNA diagnostic signature

The methodological framework of this study is structured into three sequential phases: discovery, hold-out validation, and independent validation. Initially, samples from the public database were randomly allocated, with 70% designated for the discovery phase to facilitate model training and the remaining 30% reserved for the hold-out validation phase to assess model performance. Stratified splitting was employed to maintain an equivalent ratio of tumor to control samples as in the original dataset, thereby mitigating bias arising from sample imbalance. Plasma exosome samples procured from RUSH University Medical Center were exclusively applied to the independent validation phase to substantiate the liquid biopsy's diagnostic potential.

In order to guarantee statistical robustness, the Hanley & McNeil formula for AUC-based analysis was used to determine the required sample size for evaluating the tRNA-based biomarker, taking performance validation and feature selection stability into consideration. Assuming an AUC of 0.75 in the discovery phase and 0.80 in the validation phase, with 95% confidence ($\alpha = 0.05$), 80% power ($\beta = 0.20$), and a null hypothesis AUC of 0.5, the minimum required sample sizes were ~90 patients (72 cases, 18 controls, r = 4.3) for discovery, ~44 patients (36 cases, 8 controls, r = 4.3) for hold-out validation, and ~72 patients (36 cases, 36 controls, r = 1.0) for independent validation. Our study exceeded these estimates, including 1012 patients in discovery (821 cases, 191 controls, r = 4.3), 434 in hold-out validation (352 cases, 82 controls, r = 4.3), and 202 in independent validation (117 cases, 116 controls, r = 1.0), ensuring high statistical power.

Before conducting our analysis, we performed batch effect correction on the stratified data from different phases using the limma package (version 3.56.2) in R, preparing it for meta-analytical processing. To identify the tRNA signature with the highest diagnostic value, we utilized the MetaIntegrator package (version 2.1.3, https://github.com/cran/MetaIntegrator) in R, which consolidates effect sizes and statistical significance of gene expression across multiple studies[24]. This approach facilitates the detection of significant tRNAs demonstrating meaningful effect sizes. Our study adopted stringent filtering criteria: (1) a minimum meta-effect size threshold of 0.2 with a False Discovery Rate (FDR) not exceeding 30% for the combined effect sizes; (2) integration of gene significances through Fisher's test, requiring an FDR below 0.001; (3) analysis across all six datasets included in the study.

Additionally, our methodology incorporates a leave-one-out validation algorithm from the MetaIntegrator package (version 2.1.3), which systematically excludes one dataset at a time to conduct meta-analysis on the remainder. This iterative approach ensures that each dataset is removed once, enabling the identification of a robust tRNA signature resilient to potential biases associated with any single dataset.

The final optimization of the gene set was achieved using the forwardSearch and backwardSearch algorithms, which are iterative, greedy-based approaches designed to maximize the predictive performance of the tRNA signature. The forwardSearch algorithm begins with a single tRNA and incrementally adds additional tRNAs from the candidate pool, each selected for its ability to enhance the weighted AUC. Weighted AUC is calculated as the sum of the AUCs for individual datasets, each adjusted by sample size, to prioritize performance across the largest sample coverage. This addition process continues until further tRNA inclusion no longer improves the weighted AUC. In contrast, the backwardSearch algorithm begins with the entire collection of potential tRNAs and systematically eliminates specific tRNAs whose removal results in the greatest rise in weighted AUC. This elimination process continues until no further eliminations improve the weighted AUC.

Using the intersection of results from both algorithms produced a robust six-tRNA signature, ensuring a highly optimized selection with minimized redundancy and maximized discriminatory power across datasets. This six-tRNA signature was then validated through ROC and PRC analysis, combining the T-scores of the selected tRNAs to confirm its diagnostic accuracy across multiple phases and datasets.

Clinical data were accessible during data preprocessing and statistical analyses, including patient demographics and clinical status. Therefore, rather than directly selecting features based on clinical labeling, model construction was done in a data-driven manner to reduce potential bias. ROC and PRC analyses were used in the validation phase to evaluate the identified signature's diagnostic performance with known cancer/control classifications.

## Assessing robustness and diagnostic efficacy of the tRNA signature

The standardized tRNA signature score (T-score) of each sample was computed to classify samples as oncogenic or non-oncogenic. The T-score calculation relied on the expression levels (TPM-normalized) of the six tRNA signature genes across all samples in the dataset. Specifically, the calculation involves three steps:

1) Geometric Mean Calculation:
The geometric means of up-regulated ($S_{up}$) and down-regulated ($S_{down}$) tRNAs for each sample $i$ were calculated as:

$$GeoMean_{up}^{i} = \left( \prod_{j \in S_{up}} TPM_j^i \right)^{\frac{1}{|S_{up}|}} \tag{1}$$

$$GeoMean_{down}^{i} = \left( \prod_{j \in S_{down}} TPM_j^i \right)^{\frac{1}{|S_{down}|}} \tag{2}$$

2) Preliminary Score: The preliminary score was computed as:

$$Score^i = GeoMean_{up}^i - GeoMean_{down}^i \tag{3}$$

3) Standardization:

$$T - Score^i = \frac{Score^i - \mu}{\sigma} \tag{4}$$

where $\mu$ and $\sigma$ are the mean and standard deviation of all preliminary scores.

To evaluate the discriminatory efficacy of the tRNA signature between cancerous and non-cancerous specimens, ROC and PRC analyses were employed based on the T-scores of samples across different phases. Herein, the AUC and the Area Under the Precision-Recall Curve (AUPRC) were determined, respectively, and summarized by the sample size weight. In the independent validation phase, the signature's diagnostic accuracy was

further scrutinized, contrasting NSCLC with benign and normal samples. Concurrently, the Mann–Whitney U test was utilized to compare T-scores between cancer and control samples across various studies. Similarly, the robustness of the tRNA signature was verified through distinct sample segregations based on age, gender, racial group, cancer subtype, smoking history, and histological stage. We systematically analyzed all available demographic and clinical data across datasets, with missing values excluded from the analysis (labeled as "NA" in Table 1). The results were visualized using the summaryROCPlot and multiplePRCPlot functions from the MetaIntegrator package (version 2.1.3), along with pROC (version 1.18.5) and ggplot2 packages (version 3.4.4) in R.

### Survival analysis

We conducted survival analysis using retrospective clinical data (vital status, follow-up duration) from the TCGA-LUAD and TCGA-LUSC cohorts (2004–2013), combined with TPM-normalized tRNA expression data. The clinical endpoints included overall survival, defined as the time from diagnosis to death from any cause; and progression-free survival, defined as the time from diagnosis to disease progression or death, whichever occurred first. Patients with incomplete follow-up data or those lost to follow-up before a survival event were excluded, and samples with low expression in over 50% of tRNAs were removed to ensure robust biomarker evaluation.

The required sample size for survival analysis was initially estimated using Schoenfeld's formula for Cox proportional hazards regression, assuming HR = 1.5, a 60% event rate, 80% power, and $\alpha = 0.05$, yielding a minimum of 80 patients (48 observed death events). To maintain statistical power, we included the full available dataset of 976 NSCLC patients without pre-stratification.

We developed a risk score based on the expression levels of tRNAs, assigning scores of 1 for high expression and 0 for low, and determined a risk score cut-off using the Survminer package (version 0.4.9). Kaplan-Meier curves visualized survival implications. A multivariate Cox regression model incorporating risk scores with patient demographics and clinical features (age, cancer subtype, sex, smoking history, AJCC cancer stage) was used to evaluate prognostic factors. The results were visualized with ggplot2 (version 3.4.4), highlighting the prognostic relevance of tRNA in lung cancer.

For the analysis of pathway-specific survival, Z-scores for pathway activity were computed by aggregating the TPM-normalized RNA expression values of pathway-associated genes identified through RNAhybrid (version 2.1.2) and KEGG enrichment analysis (using a significance threshold of $p < 0.05$). A Cox proportional hazards model was employed to calculate risk scores for each sample based on pathway activity. The median risk score was used to categorize samples into high-risk and low-risk groups. Kaplan-Meier survival analysis was then conducted for overall metabolic pathways and individual pathways.

After completing all survival analyses, a post-hoc power analysis was conducted to evaluate the statistical power of the study based on the observed hazard ratio (HR) and actual number of survival events. While the initial sample size estimation was based on HR = 1.5, the observed HR ranged from 1.3 to 2.2, prompting a reassessment of statistical adequacy. With 976 NSCLC patients and 390 observed deaths, the post-hoc power remained high ( ~ 95.6% to ~100%), confirming that the study is sufficiently powered to detect the moderate to strong prognostic impact of the tRNA signature.

### Exploration of tRF expression data

We adopted limma package (version 3.56.2) to correct the batch effects across various datasets on TPM-normalized tRF expression data. Subsequently, the corrected data underwent meta-analysis using the MetaIntegrator package (version 2.1.3). This analysis focused on identifying tRFs with significant differential expression between cancerous and non-cancerous samples, characterized by a substantial effect size and an FDR of less than 0.05.

### Correlation analysis between signature tRNAs and their originated tRFs

A Spearman rank correlation analysis was conducted to investigate the relationship between the expression levels of six signature tRNAs and their corresponding tRFs, utilizing TPM-normalized expression data for both tRNAs and tRFs. This analysis was performed on expression data from both individual datasets and a combined dataset comprising all tissue samples. The analysis outcomes, whether derived from individual or aggregated datasets, were visualized using the pheatmap package in R (version 1.0.12).

### tRF-targeted gene identification and functional enrichment

RNAhybrid (version 2.1.2) was utilized on a Linux platform to predict the target genes of two tRF biomarkers derived from our identified signature tRNAs. Target sequences for hybridization were obtained from human mRNA FASTA files (GRCh38/hg38) sourced from GENCODE Human (https://www.gencodegenes.org/human/). The criteria for prediction included setting the minimum free energy (MFE) threshold for hybridization between two sequences at −20 kcal/mol and applying a pvalue cutoff of 0.05. This analysis resulted in the identification of 712 potential target genes for the tRFs.

We performed functional enrichment analyses on the predicted target genes using KEGG pathway and Gene Ontology (GO) analysis. KEGG pathways were analyzed with the NIH DAVID Functional Annotation Tool (https://david.ncifcrf.gov), while GO terms were analyzed using the online tools of the Gene Ontology Resource (https://www.geneontology.org/). Significantly enriched pathways ($p < 0.05$) and GO terms (FDR < 0.05) were visualized using ggplot2 (version 3.4.4) in R and Cytoscape (version 3.10.1).

### Statistics and reproducibility

Small RNA sequencing data were obtained from GEO (GSE110907, GSE62182, GSE83527, and GSE175462) and TCGA (TCGA-LUAD and TCGA-LUSC), totaling 1446 lung tissue samples (1173 tumor, 273 normal) and 233 plasma samples from RUSH University Medical Center for independent validation. Upstream analysis included FastQC (version 0.11.9), fastp (version 0.20.0), STAR (version 2.7.10), and FeatureCounts (version 2.0.6), with limma (version 3.56.2) applied for batch effect correction. Statistical analyses were performed in R (version 3.4.4) using standardized bioinformatics workflows. MetaIntegrator (version 2.1.3) identified a six-tRNA diagnostic signature, validated via leave-one-out cross-validation, forwardSearch/backwardSearch optimization, and ROC/PRC analysis. Kaplan-Meier survival analysis and multivariate Cox regression evaluated prognostic relevance, while Spearman correlation examined tRF-tRNA relationships. Functional analysis was conducted using RNAhybrid (version 2.1.2), KEGG (https://david.ncifcrf.gov), and GO enrichment (https://www.geneontology.org/), ensuring robustness and reproducibility.

### Reporting summary

Further information on research design is available in the Nature Portfolio Reporting Summary linked to this article.

## Results

### tRNAs emerge as promising candidates for diagnostic signatures in lung cancer detection

A comprehensive analysis was conducted on a substantial dataset comprising 1679 samples, including 1446 tissue samples sourced from six distinct public datasets (GSE110907, GSE62182, GSE83527, GSE175462, TCGA-LUAD, and TCGA-LUSC), as well as 233 exosome samples obtained from RUSH University Medical Center (Fig. 1a). The sample distribution across cancer and control groups (including benign and normal samples) was nearly balanced across datasets, except for TCGA, which predominantly consisted of 991 cancer samples and 91 normal samples. Overall, the study encompassed 1290 cancer samples, 335 normal samples, and 54 benign samples, with detailed demographics, cancer subtypes, the American Joint Committee on Cancer (AJCC) pathologic stages, and smoking history presented in Table 1.

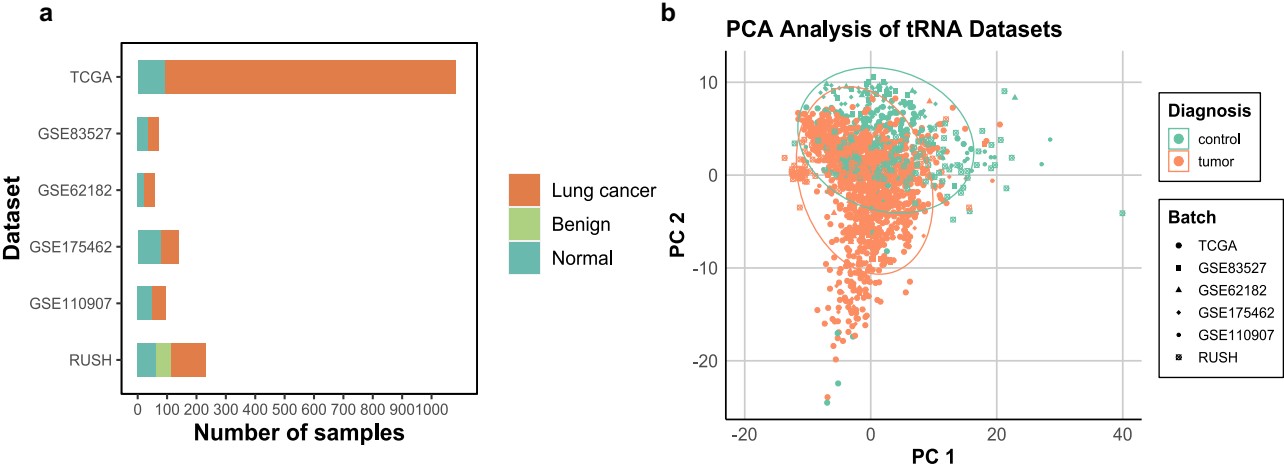

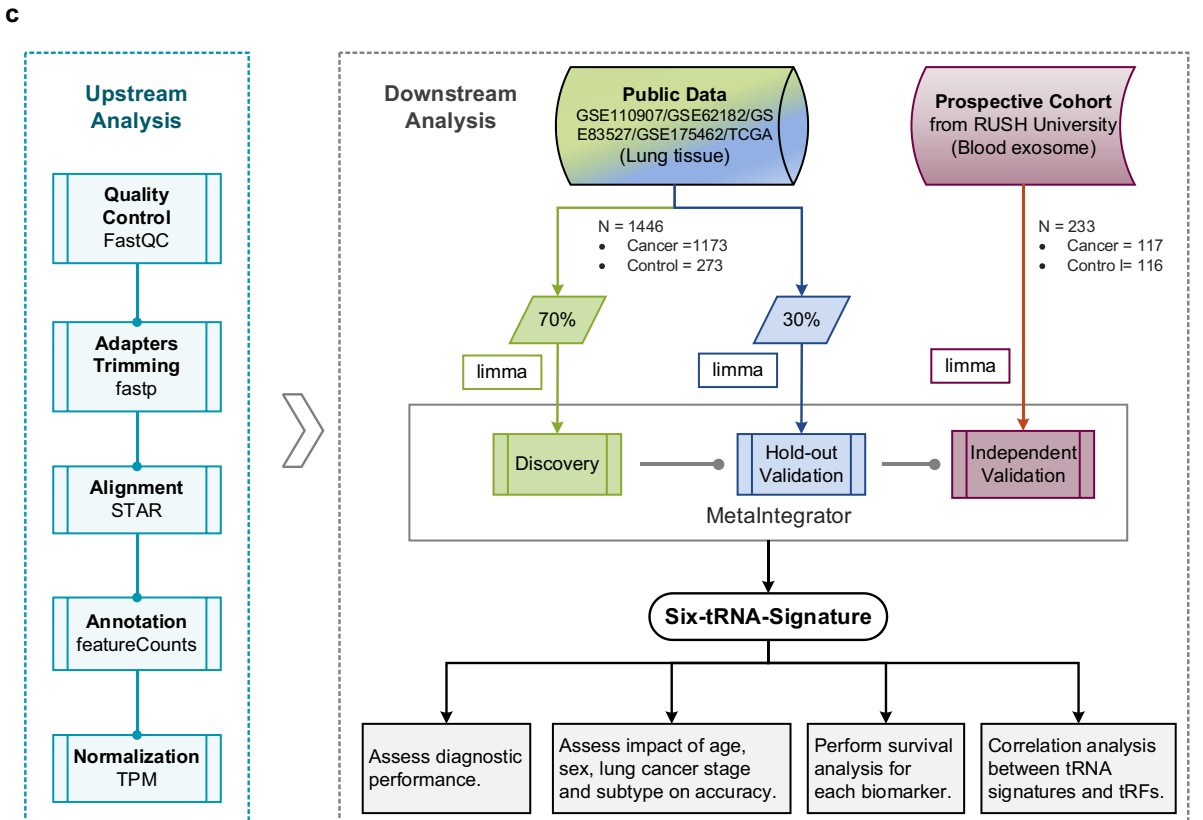

**Fig. 1 | Principal component analysis (PCA) and study design based on sample distribution. a** Sample size and diagnostic classification for each dataset included in the study. Bar colors represent different diagnostic categories. **b** PCA plot of all samples, where each point represents a specimen. Green points indicate control samples (normal and benign), orange points represent cancer samples. The shape of each point corresponds to its dataset of origin. **c** Schematic representation of the study's methodological framework.

Principal Component Analysis (PCA) revealed a significant separation in tRNA expression profiles between control and tumor groups (MANOVA, $p = 2.2 \times 10^{-16}$, partial $\eta^2 = 0.23$), highlighting the potential of tRNAs as discriminative markers for cancer detection (Fig. 1b). Furthermore, minimal expression divergence was observed between LUAD and LUSC subtypes (Fig. S2), suggesting that both NSCLC subtypes may share similar oncogenic pathways and cellular processes critical for tumor growth and survival, leading to overlapping molecular signatures.

## A six-tRNA signature has been identified and rigorously validated across multiple phases

The signature identification model was constructed using the MetaIntegrator algorithm, which integrates gene significance and effect size, supplemented by leave-one-out analysis to confirm the robustness of the identified markers[24]. To ensure the model's reliability, tissue samples were stratified into 70% for the discovery phase and 30% for hold-out validation, maintaining the original cancer-to-non-cancer ratio. Independent exosome samples were utilized for external validation (Fig. 1c).

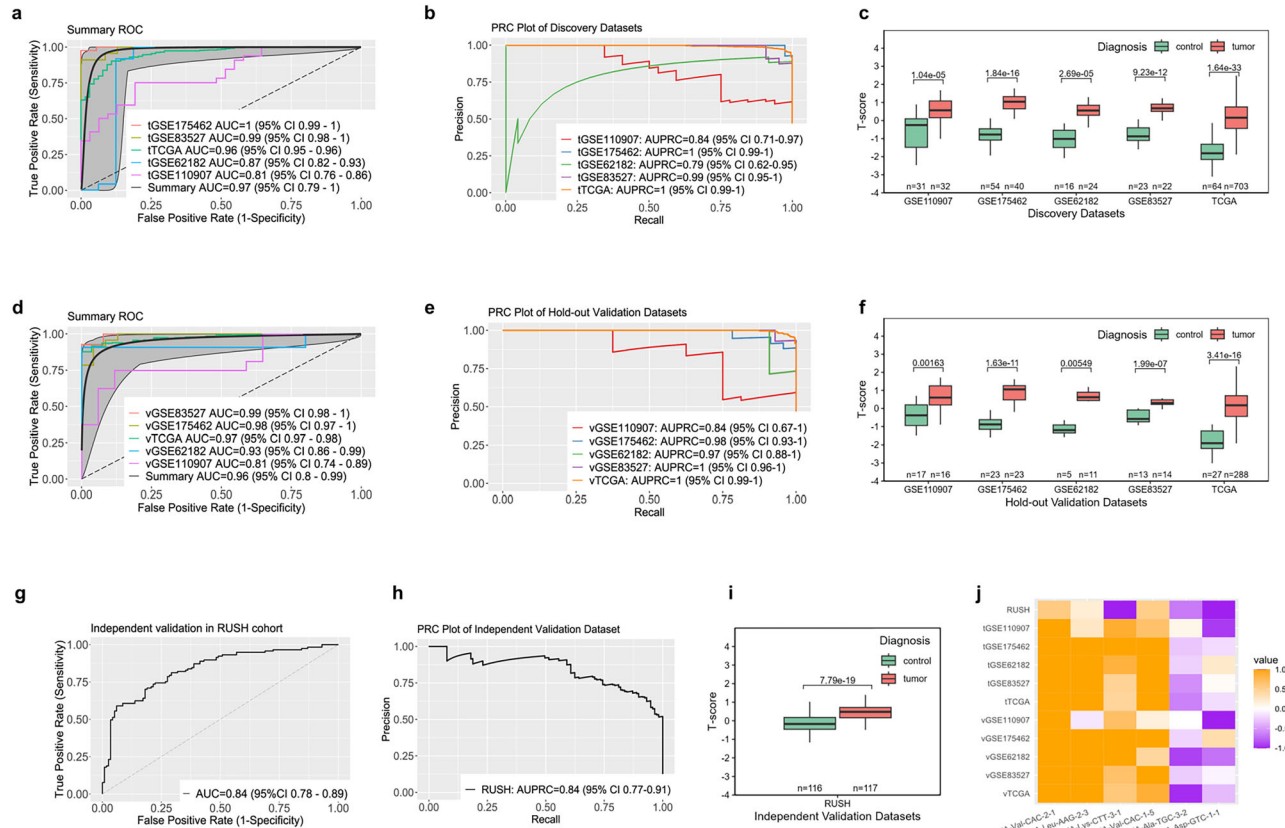

**Fig. 2 | Performance evaluation of the 6-tRNA signature across different phases.**
**a**, **d**, **g** Receiver Operating Characteristic (ROC) curves for T-scores in the discovery, hold-out validation, and independent validation phases, respectively. Curve colors correspond to their respective datasets. AUC, area under the curve; CI, confidence interval. **b**, **e**, **h** Precision-Recall Curves (PRC) for T-scores in the discovery, hold-out validation, and independent validation phases, respectively. Curve colors correspond to their respective datasets. AUPRC, area under the precision-recall curve; CI, confidence interval. **c**, **f**, **i** Boxplots of T-scores across different datasets, comparing tumor and control cohorts during the discovery, hold-out validation, and independent validation phases. Box colors represent different diagnostic categories. Sample sizes (n) for each cohort are displayed below the corresponding boxplots. Statistical comparisons were performed using the Mann–Whitney U test. Exact p-values are displayed above each comparison bar. **j** Heatmap of T-scores for each tRNA biomarker across all datasets and phases. "t" represents the discovery phase, "v" represents the hold-out validation phase, and "RUSH" denotes the independent validation phase. The color scale represents T-score values, ranging from 1.0 (orange) to -1.0 (purple).

During the discovery phase, 117 tRNAs (68 up-regulated and 49 down-regulated) were initially filtered (Data S2). Further refinement using the forwardSearch and backwardSearch algorithms led to the selection of a robust six-tRNA signature, consisting of four up-regulated tRNAs (*tRNA-Val-CAC-2-1*, *tRNA-Leu-AAG-2-3*, *tRNA-Val-CAC-1-5*, and *tRNA-Lys-CTT-3-1*) and two down-regulated tRNAs (*tRNA-Ala-TGC-3-2* and *tRNA-Asp-GTC-1-1*) in tumor samples compared to the control group.

The performance of the six-tRNA signature was evaluated using ROC and PRC curves based on the T-score, demonstrating an outstanding AUC of 0.97 in the discovery phase (Fig. 2a). PRC analysis also showed excellent results, with high AUPRCs of 0.79-1.00 across databases (Fig. 2b). Significant differences in T-scores between tumor and control samples were confirmed across datasets using Mann–Whitney U tests (Fig. 2c). The signature was then successfully validated in the hold-out validation phase using the remaining tissue samples, yielding an AUC of 0.96, with a minimal AUPRC of 0.84, and statistically significant differences between tumor and control groups (Fig. 2d–f).

Further independent validation in a cohort from RUSH University Medical Center confirmed the diagnostic potential of the six-tRNA signature, with an AUC of 0.84 and AUPRC of 0.84, along with statistically significant T-scores (Fig. 2g–i). This independent validation highlights the signature's potential for application in liquid biopsy settings. Besides, additional analyses comparing NSCLC patients with benign individuals and with healthy controls within the RUSH cohort showed high AUC values of

0.85 and 0.82, respectively, with significant distinctions in T-scores between cancerous and non-cancerous cohorts (Fig. S3). These findings reinforce the diagnostic precision of the six-tRNA signature in liquid biopsies, particularly in differentiating between malignant and benign conditions.

To confirm that the imbalanced sample sizes of cancerous and non-cancerous cases did not introduce bias during model training and validation, a stratified sampling approach was applied to the TCGA dataset. The 991 tumor samples were randomly split into ten subsets, nine of which contained 99 samples each, and one contained 100 samples. Each tumor subset was paired with 91 control samples to form ten balanced groups. The performance of the six-tRNA signature was evaluated using ROC analysis for each group, resulting in a consistent AUC of 0.95 across all groups, further validating the robustness of the signature (Fig. S4).

In conclusion, the six-tRNA signature has been successfully identified and validated across multiple phases and datasets (Fig. 2j, Fig. S5), highlighting its potential as a reliable signature for the detection of NSCLC.

## The robustness of the T-score has been substantiated across a spectrum of demographic factors, histological subtypes, smoking statuses, and AJCC pathologic stages

The collected data encompassing age, sex, histological subtype, AJCC pathologic stage, and smoking history of the study cohort are systematically presented in Table 1. The robustness of the six-tRNA signature's T-score across diverse categorizations of the samples was assessed by employing ROC, PRC analyses, and Mann–Whitney U tests (Fig. 3, S6). Integration of

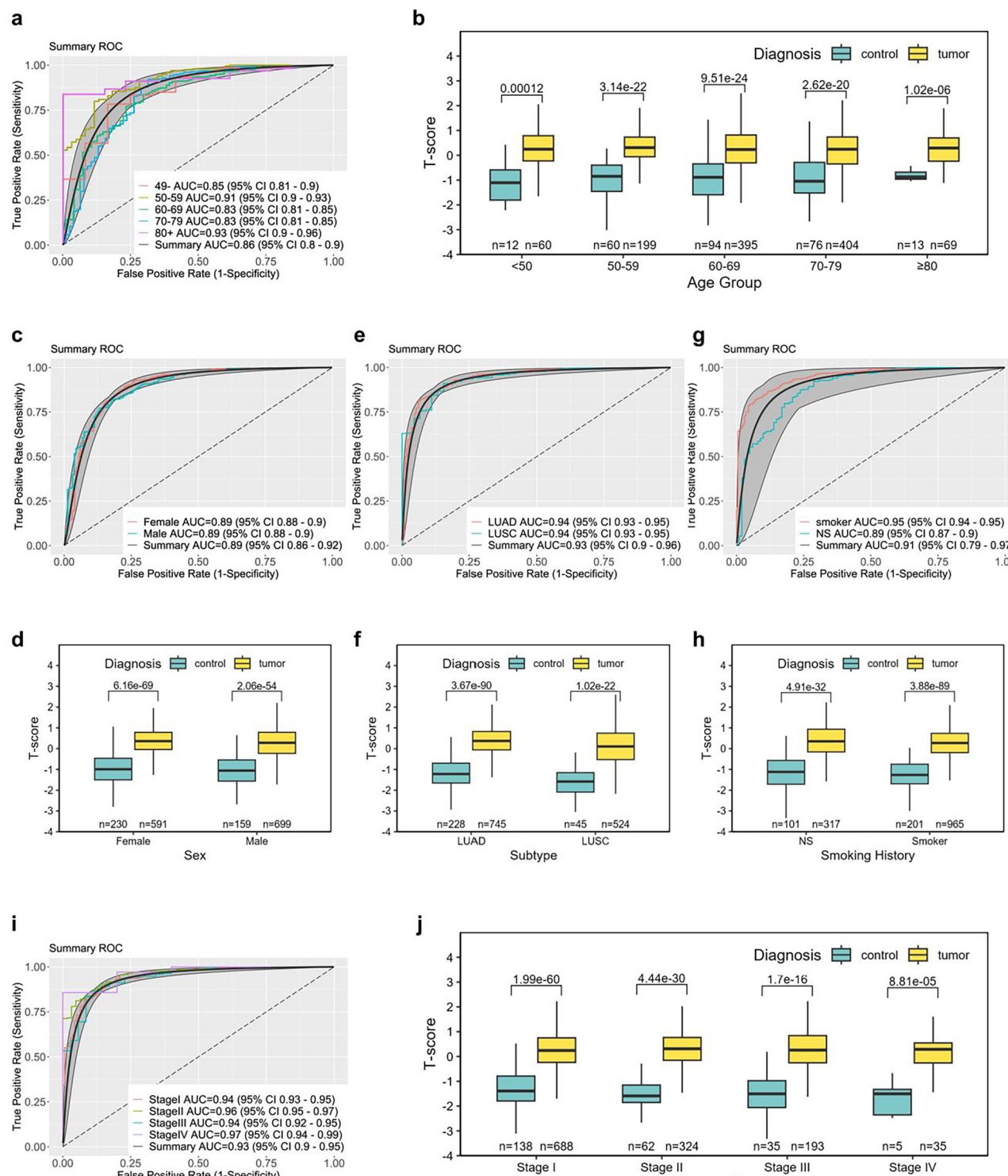

**Fig. 3 | Evaluation of T-score robustness in stratified analyses by age, sex, histological subtype, smoking history, and AJCC cancer stage in NSCLC patients.** **a**, **c**, **e**, **g**, **i** Receiver Operating Characteristic (ROC) curves assessing the discriminatory performance of the tRNA signature across different age groups, sexes, histological subtypes, smoking history categories, and AJCC cancer stages. Curve colors correspond to their respective demographic categories. AUC area under the curve, CI confidence interval. **b**, **d**, **f**, **h**, **j** Boxplots comparing T-scores between tumor and control cohorts across different stratified factors. Box colors represent different diagnostic categories. Sample sizes (n) for each cohort are displayed below the corresponding boxplots. Statistical comparisons were performed using the Mann–Whitney U test. Exact *p* values are displayed above each comparison bar.

patient age data from both TCGA and RUSH datasets revealed that the T-score consistently exhibited high performance across various age demographics, as evidenced by an AUC value of 0.86, with no individual age group demonstrating an AUC of below 0.83 (Fig. 3a). Concurrently,

statistical evaluations indicated a significant T-score difference between tumor and control samples across all age ranges (Fig. 3b). Further validation of the T-score's robustness was conducted across different sexes, histological subtypes, and smoking statuses, synthesizing data from all available datasets,

which suggests a uniformly high efficacy irrespective of these variables (Fig. 3c–h). Crucially, comparative analyses of the T-score's effectiveness across various AJCC pathologic stages revealed its remarkable proficiency in diagnosing both early-stage (stages I and II) and advanced-stage (stages III and IV) NSCLC, with each stage demonstrating AUC values of no less than 0.94 (Fig. 3i). Correspondingly, a significant distinction was observed between the tumor and control samples, underscoring the T-score's accurate diagnostic performance for early-stage NSCLC (Fig. 3j). PRC analyses across these factors also demonstrated outstanding performance, with all AUPRC values exceeding 0.90, further supporting the diagnostic robustness of the 6-tRNA signature (Fig. S6).

In addition to these factors, the T-score was further validated across racial groups. The analysis revealed that the T-score demonstrated excellent diagnostic performance in White and Asian populations, with AUC values exceeding 0.84. However, its diagnostic accuracy was notably lower in Black individuals, reflecting a reduced AUC and diminished statistical significance. This discrepancy may be due to the underrepresentation of Black individuals in the datasets, which limits the model's ability to capture potential biological variations specific to this population (Fig. S7).

In conclusion, the T-score emerges as a potential independent biomarker for NSCLC diagnosis, transcending variations in sex, age groups, histological subtypes, smoking history, and pathologic stages.

### Three tRNAs within the signature were validated to have prognostic values that are capable of being independent predictive factors

Upon analyzing data from 976 lung cancer patients (filtered from the original 1,089 cases in the TCGA repository, including 585 TCGA-LUAD and 504 TCGA-LUSC samples), three tRNAs from the identified six-tRNA signature—*tRNA-Lys-CTT-3-1*, *tRNA-Val-CAC-2-1*, and *tRNA-Leu-AAG-2-3*—demonstrated significant prognostic value. Kaplan-Meier survival analysis, followed by log-rank testing, revealed that patients with higher expression levels of these tRNAs exhibited significantly poorer survival outcomes. Specifically, *tRNA-Lys-CTT-3-1* was associated with a hazard ratio (HR) of 1.37 ($p = 2.44 \times 10^{-3}$), *tRNA-Val-CAC-2-1* with an HR of 1.33 ($p = 3.02 \times 10^{-2}$), and *tRNA-Leu-AAG-2-3* with an HR of 1.30 ($p = 1.07 \times 10^{-2}$). These findings suggest that elevated expression of these three tRNAs is linked to a higher risk of mortality, highlighting their potential as independent prognostic biomarkers in lung cancer (Fig. 4a–c).

To further assess the prognostic significance of these biomarkers, a cumulative risk score was derived by integrating the expression levels of *tRNA-Lys-CTT-3-1*, *tRNA-Val-CAC-2-1*, and *tRNA-Leu-AAG-2-3*. Patients were categorized into high-risk and low-risk groups based on an optimal threshold identified using the Survminer package. Kaplan-Meier survival analysis revealed a statistically significant survival disparity between the two

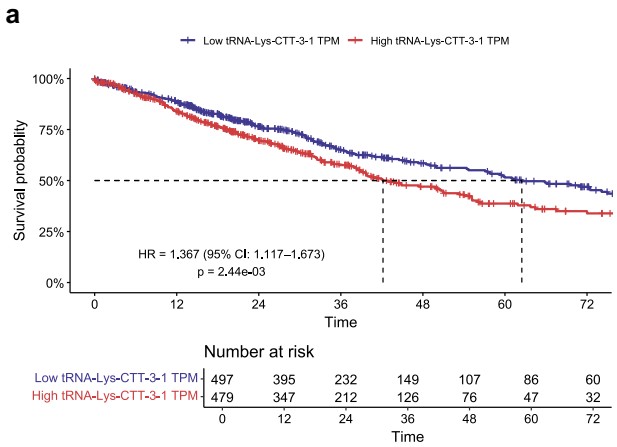

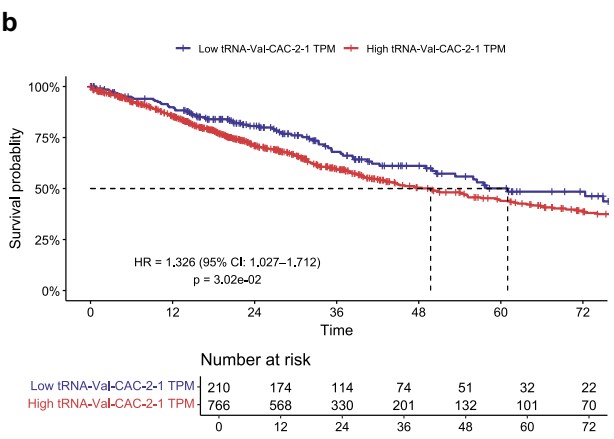

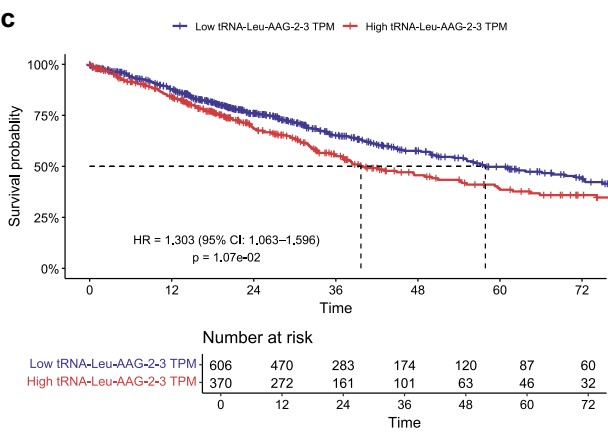

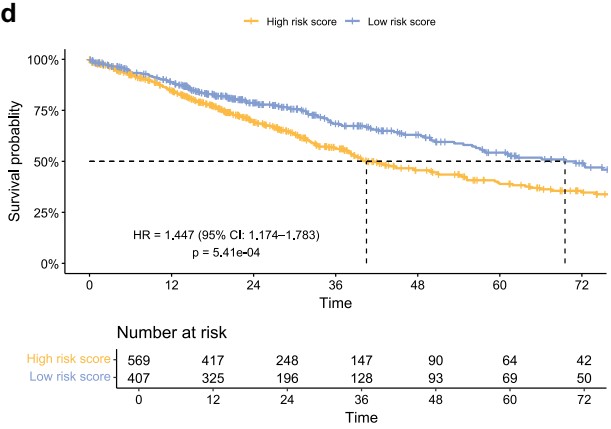

**Fig. 4 | Survival analysis of individual tRNA biomarkers and composite risk scores in the TCGA patient cohort. a–c** Kaplan-Meier survival curves illustrating the prognostic impact of individual tRNA biomarkers in NSCLC patients. Curve colors correspond to expression levels, where blue represents low expression and red represents high expression. **d** Kaplan-Meier survival curve showing the effect of the composite risk score on patient prognosis. Curve colors correspond to risk score groups, with yellow indicating the high-risk group and blue indicating the low-risk group. Dashed lines denote the median survival threshold, corresponding to a 50% survival probability for the respective patient cohorts at specific time points. The tables below each plot indicate the number of patients at risk over the 72-month observation period. Statistical significance was determined using the log-rank test. HR hazard ratio, CI confidence interval.

**Table 2 | Multivariate cox regression analysis of independent predictors of survival in NSCLC patients**

| Subgroup | Number | Adjusted HR (95% CI) | P value | Significance |
|---|---|---|---|---|
| **Risk_score** | | | | |
| Low | 415 | Reference | Reference | |
| High | 576 | 1.80 (1.45–1.17) | 6.27E-04 | *** |
| **Subtype** | | | | |
| LUAD | 513 | Reference | Reference | |
| LUSC | 478 | 1.08 (0.86–1.35) | 5.15E-01 | |
| **Sex** | | | | |
| Female | 398 | Reference | Reference | |
| Male | 593 | 1.03 (0.83–1.29) | 7.84E-01 | |
| **Age** | | | | |
| Age | 991 | 1.01 (1.00–1.03) | 1.89E-02 | * |
| **Smoking_history** | | | | |
| No | 235 | Reference | Reference | |
| Yes | 756 | 0.88 (0.69–1.13) | 3.25E-01 | |
| **AJCC_stage** | | | | |
| Stage I | 507 | Reference | Reference | |
| Stage II | 279 | 1.52 (1.19–1.95) | 8.56E-04 | *** |
| Stage III | 164 | 2.29 (1.76–2.97) | 7.61E-10 | *** |
| Stage IV | 30 | 3.28 (2.05–5.24) | 7.17E-07 | *** |

The reference category for each variable is indicated as "Reference" under the "Adjusted HR" and "P value" columns, serving as the baseline for comparison. Statistical significance is denoted by asterisks: "*" for $p < 0.05$; "**" for $p < 0.01$; "***" for $p < 0.001$.

groups (log-rank test, $p = 5.41 \times 10^{-4}$, HR = 1.44), with low-risk patients exhibiting a median survival advantage of approximately three years over those in the high-risk group during the 72-month follow-up period. Notably, the cumulative risk score demonstrated superior prognostic performance, providing greater predictive accuracy than any single tRNA alone (Fig. 4d).

Additional validation using multivariate Cox regression analysis confirmed the risk score as an independent prognostic factor for NSCLC, alongside other factors such as pathological subtype, sex, age, smoking history, and AJCC cancer stage (Wald test, $p = 6.27 \times 10^{-4}$, HR = 1.80; Table 2). Similar to previous findings, the lower risk score predicted longer survival, underscoring its potential as an efficient prognostic tool in NSCLC[18].

**Two tRNAs in the signature may exert regulatory influences through associated tRNA fragments**

Transfer RNA fragments (tRFs) have emerged as vital regulators in the post-transcriptional process of tumorigenesis through interactions with mRNA molecules[25]. Despite their recognized roles, the connection between tRFs and their parent tRNA expressions has not been well-characterized. To address this gap, the study investigated the expression patterns correlating tRFs with identified tRNAs. We found 69 tRFs with substantial differential expression between malignant and non-cancerous samples: 18 were up-regulated, and 51 were down-regulated (Fig. 5a, Data S3). Intriguingly, two up-regulated tRFs, namely 5'-tRF-Val-CAC-2-1 and 5'-tRF-Val-CAC-1-5, were derived from signature tRNAs (Fig. 5b).

Based on TPM-normalized expression profiles, a Spearman correlation analysis was performed across individual datasets and aggregate samples to see whether there was a correlation between these tRFs and their source tRNAs. 5'-tRF-Val-CAC-2-1 / 5'-tRF-Val-CAC-1-5 and their parent tRNAs (tRNA-Val-CAC-2-1 / tRNA-Val-CAC-1-5) showed a strong positive association with each other, with correlation coefficients of 0.78 and 0.75 in the entire sample population, respectively (Fig. 5c). Individual datasets consistently showed this link (Fig. S8), suggesting that the signature tRNAs may use the corresponding tRFs to achieve their regulatory function.

Following these results, a strict minimum free energy (MFE) criterion of -20 kcal/mol was applied to the RNAhybrid tool to discover mRNA targets that might interact with the identified tRFs. A total of 957 mRNAs were found to interact with the tRFs in a significant way in our analysis (Data S4). Significant enrichment was found in six important pathways by subsequent KEGG pathway enrichment analysis, with metabolic pathways being the most significantly enriched, comprising 92 identified genes. Other enriched pathways included glycan degradation, homologous recombination, RNA polymerase, adherens junction, and O-glycan biosynthesis. These pathways are crucial for preserving the genomic stability and energy production needed for tumor cell survival and proliferation (Fig. 5d, e; Data S5)[26–30].

Similarly, Gene Ontology (GO) enrichment analysis identified important gene accumulations in biological processes. According to gene count, the top five processes were metabolic, organic substance metabolic, cellular, and main metabolic processes (Fig. 5f; Data S6). These fundamental cellular and metabolic processes are often altered in cancer by modifying cellular metabolism and other processes to favor tumorigenesis[31,32]. Furthermore, cellular elements such as organelles and intracellular structures were noticeably overrepresented (Fig. 5g), and molecular function analysis revealed gene aggregation linked to binding and catalytic activities (Fig. 5h), supporting the various functions of tRFs in cellular function and cancer development.

To explore the potential regulatory role of tRFs and their target pathways, we conducted survival analysis on the expression levels of specific metabolic pathways (Fig.6, S9). Kaplan-Meier survival analysis for the overall metabolic pathway demonstrated significant prognostic value in NSCLC (log-rank test, $p = 4.26 \times 10^{-15}$, HR = 2.29; Fig. 6a). KEGG analysis of these tRF target genes highlighted pathways including biosynthesis of cofactors, purine metabolism, oxidative phosphorylation, other types of O-glycan biosynthesis, nucleotide metabolism, lysine degradation, amino acid biosynthesis, and the phosphatidylinositol signaling system (Fig. 6b, Data S7). Further survival analysis revealed significant prognostic differences across multiple metabolic pathways, including biosynthesis of cofactors ($p = 4.17 \times 10^{-5}$, HR = 1.52; Fig. 6c), purine metabolism ($p = 3.21 \times 10^{-2}$, HR = 1.24; Fig. 6d), oxidative phosphorylation ($p = 3.98 \times 10^{-2}$, HR = 1.23; Fig. 6e), and alanine, aspartate, and glutamate metabolism ($p = 2.06 \times 10^{-4}$, HR = 1.46; Fig. 6f). In all cases, lower risk scores correlated with prolonged survival, underscoring a strong association between these metabolic pathways and NSCLC progression.

In conclusion, our results suggest that tRNAs, through their processing into tRFs and their influence on mRNA stability and translational regulation, play an essential role in modulating metabolic networks crucial for NSCLC tumorigenesis.

## Discussion

This study developed a comprehensive model to analyze small RNA sequencing data from multiple studies, identifying a six-tRNA signature for NSCLC diagnosis. Utilizing 1446 lung tissue samples from five public datasets, the study proceeded through an initial discovery phase for rigorous model training, followed by a hold-out validation phase to confirm the model's robustness and the efficacy of the gene signature. Independent validation using a cohort of 233 plasma exosome samples further

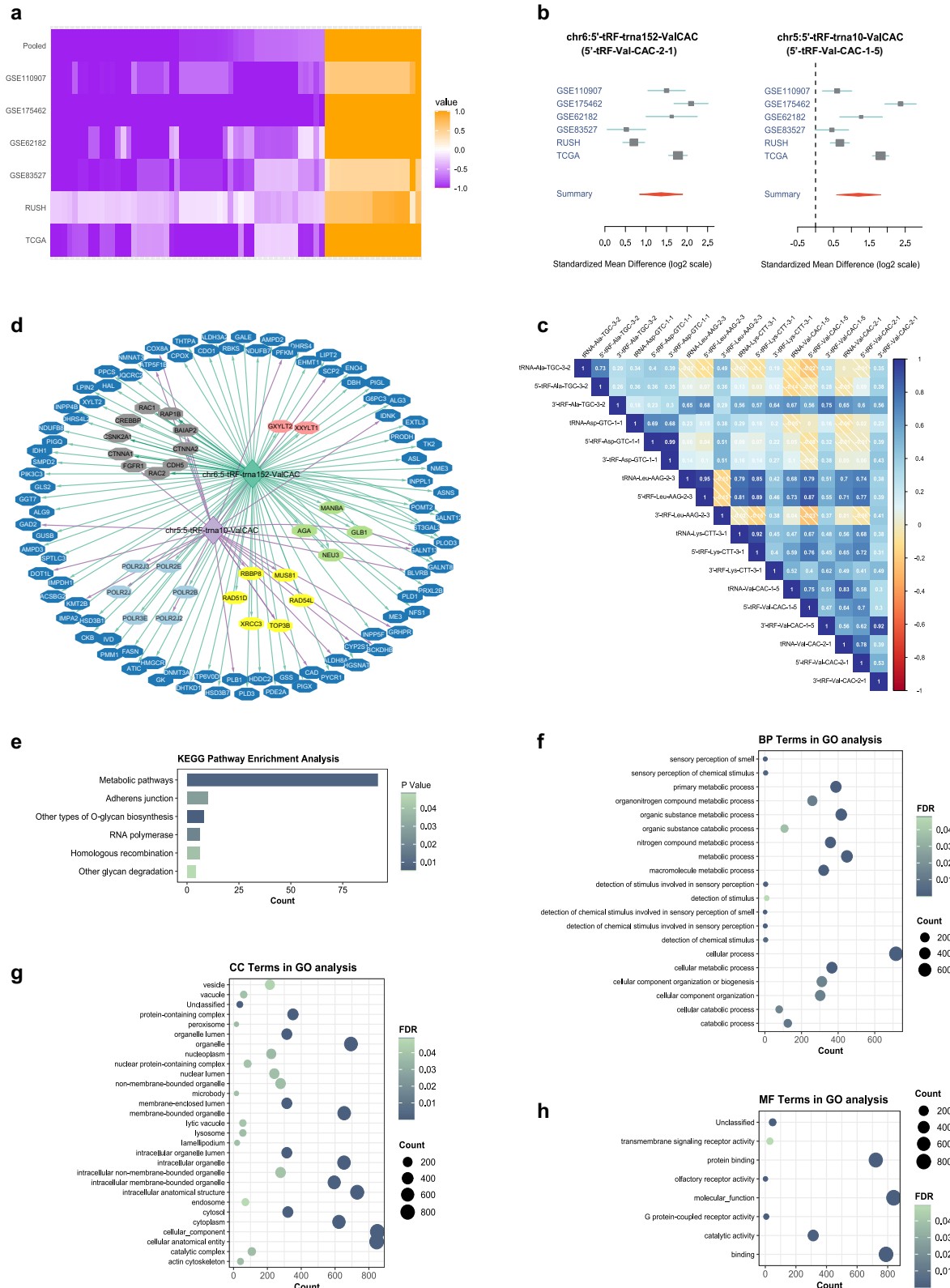

substantiated the diagnostic accuracy of the six-tRNA signature, demonstrating its utility for non-invasive, real-time liquid biopsies. Notably, the diagnostic performance of the signature remained consistently high across various demographics and clinical characteristics, including age, gender, smoking history, cancer subtypes, and histological stages. Furthermore, the signature's capacity to distinguish between cancerous and benign conditions

highlights its noteworthy potential for clinical application in the early screening of NSCLC.

tRNAs are classified into three distinct hierarchical levels: isoacceptors, isodecoders, and individual tRNAs, with isoacceptors comprising tRNAs with differing anticodons but carrying the same amino acid, isodecoders sharing identical anticodons but differing in nucleotide sequences, and

**Fig. 5 | Regulatory impact of signature tRNAs through associated tRNA fragments (tRFs). a** Heatmap illustrating the expression levels of differentially expressed tRFs across multiple datasets. The color scale represents T-score values, ranging from 1.0 (orange) to –1.0 (purple). **b** Forest plot depicting the meta-analysis of standardized mean differences (SMDs) for two tRFs across different datasets. Each dataset is represented by an individual effect estimate (gray box), with 95% confidence intervals (CIs) shown as blue lines. The size of the gray box reflects the sample size within each dataset. The red diamond represents the overall summary estimate from the meta-analysis. A positive SMD indicates higher expression in tumor samples, while a negative SMD suggests lower expression. **c** Heatmap of Spearman's rank correlation coefficients between signature tRNAs and their corresponding tRFs, aggregated from all datasets. The color scale represents correlation coefficients ranging from –1 (red) to 1 (dark blue). **d** Functional interaction network illustrating the relationships between two identified tRFs and their associated mRNAs, as determined by KEGG pathway enrichment analysis. Diamonds represent tRFs; ellipses denote mRNAs. The colors of mRNA groups correspond to distinct KEGG pathways. **e** Bar plot summarizing the results of KEGG pathway enrichment analysis. Bar length represents the number of genes enriched in each pathway. Statistical significance is indicated by a *p* value scale ranging from 0.05 (light green) to 0.01 (gray). **f, g, h** Bubble plots representing Gene Ontology (GO) enrichment analysis results, detailing biological processes (BP), cellular components (CC), and molecular functions (MF). Bubble size corresponds to the number of enriched genes, and false discovery rate (FDR) values are color-scaled from 0.05 (light green) to 0.01 (gray) to indicate statistical significance.

individual tRNAs maintaining constant sequences but located at different genomic sites. Historically, the focus on tRNAs within cancer research has predominantly been on the differential expression of isoacceptors and the resultant protein variations, given their classical function in amino acid transfer during mRNA translation[33]. However, recent advancements have illuminated the important roles of tRNAs in non-canonical functions, especially in epigenetics, such as tRNA fragments' regulatory effects on gene expression in post-transcription, underscoring the imperative to investigate tRNAs beyond isoacceptors to include isodecoders and individual tRNAs[34]. Our study represents an effort to identify lung cancer-specific diagnostic tRNA signature at the individual tRNA level, revealing the complex network of non-traditional tRNA functions from an epigenetic standpoint, thereby emphasizing the importance and potential of exploring the multifaceted roles of tRNAs in cancer pathogenesis.

Interestingly, our pre-analysis using PCA revealed no significant difference in tRNA expression between the two main NSCLC subtypes, LUAD and LUSC. As essential components of protein synthesis and cellular metabolism, tRNAs are often dysregulated across various cancer types. However, their expression may not differ significantly between closely related subtypes within the same organ. This observation aligns with findings from the public database DBtrend (https://trend.pmrc.re.kr/)[35], indicating no significant differences in tRNA expression between LUAD and LUSC cohorts. The absence of distinct separation between these subtypes further confirms the feasibility of studying tRNA diagnostic biomarkers across the entire NSCLC spectrum, encompassing both LUAD and LUSC.

We included all collected samples in the model development process to strengthen the robustness of our signature identification model. However, this approach presented a challenge of sample imbalance, which is mainly characterized by the greater number of cancerous samples relative to non-cancerous ones. To address this issue and mitigate potential biases in model construction, we implemented several strategies. Firstly, we opted for a stratified splitting approach instead of random sampling for dividing the samples into the discovery and hold-out validation phases, ensuring the maintenance of the original proportion of tumor samples to control samples within the dataset and reducing bias due to uneven sample distribution. Secondly, we utilized the Leave-One-Out (LOO) algorithm within the MetaIntegrator framework, which systematically excludes one dataset at a time for analysis, cycling through until each dataset has been excluded once, thus preventing the dominance of any single study due to its large sample size in influencing the outcome and ensuring that the identified signature is representative across all datasets. Thirdly, to rigorously evaluate the diagnostic performance of our model, we employed the Mann-Whitney U test in conjunction with PRC analysis; the Mann–Whitney U test determines the significance of the identified signature within each dataset for a given phase, while the PRC is more sensitive to the imbalanced samples and provides a more nuanced assessment of diagnostic performance. Lastly, a targeted examination of the TCGA dataset, the main source of sample imbalance, was conducted through the resampling of tumor samples, which were then systematically paired with normal controls, confirming the high diagnostic efficacy of the six-tRNA signature and affirming that no bias was introduced during the model construction. This multifaceted strategy demonstrates our dedication to creating an unbiased and scientifically valid diagnosis methodology.

Our six-tRNA signature, initially identified in tissue samples and subsequently validated in both tissue and plasma exosomes, demonstrates outstanding robustness. Exosomes, which can be released into the circulation from various parent cells, are actively generated by tumor cells and can accumulate to concentrations of at least $10^9$ vesicles per milliliter of blood. The abundance of exosomes secreted by tumor cells demonstrates a "seeds and plants" relationship between circulating exosomes and malignancies[36]. These exosomes act as mediators of cell-to-cell communication at distant sites, carrying functional particles—including DNA, RNA, lipids, proteins, and metabolites—originating from tumor cells and thus facilitating tumor growth and metastasis[37]. By validating the tissue-identified signature in exosomes, our results support the biomarker's ability to accurately detect the presence of NSCLC in a variety of sample types. This versatility highlights its clinical applicability, underscoring its promising potential for diagnostic applications in both invasive and non-invasive settings.

The identification of the six signature tRNAs in our study aligns with findings from broader tRNA research in humans. A previous study developed a comprehensive tRNA database (DBtRend, available at https://trend.pmrc.re.kr/) using a specialized pipeline to analyze human small RNA or microRNA sequencing data from TCGA and GEO datasets[35]. The signature tRNAs identified in our research directly correspond with their regulatory trends as DBtRend indicates. Specifically, *tRNA-Val-CAC-2-1*, *tRNA-Leu-AAG-2-3*, *tRNA-Val-CAC-1-5*, and *tRNA-Lys-CTT-3-1* demonstrated significant upregulation in tumor samples from both the TCGA-LUAD and TCGA-LUSC cohorts compared to normal controls[35]. Additionally, Kuang et al. reported that high expression of *tRNA-Leu-AAG* is associated with specific pathological subtypes of LUAD, while upregulation of *tRNA-Lys-CTT* is linked to tumor differentiation[18]. This consistency with previous research underscores the reliability of the signature tRNAs identified in our investigation.

Three distinct tRNAs—*tRNA-Lys-CTT-3-1*, *tRNA-Val-CAC-2-1*, and *tRNA-Leu-AAG-2-3*—were revealed to be significant predictors of survival in NSCLC patients, supporting the predictive value of the identified tRNA signature. The significance of *tRNA-Lys-CTT-3-1* in our findings is supported by previous research that has demonstrated its involvement in LUAD prognosis[18]. As a useful independent predictor of survival, the composite risk score derived from these tRNAs offers an improved method of patient categorization. Our findings are also comparable to a few prognostic studies, which link worse survival outcomes to advanced cancer stages and advanced age[38–40]. Furthermore, while gender and histological subtype had minor effects on survival, our data suggest a tendency for male patients to be at higher risk, which is consistent with research that predominantly attributes these differences to demographic and diagnostic factors[41,42]. Likewise, although the difference was not statistically significant between cancer subtypes, LUAD patients tended to have higher survival outcomes than LUSC patients[43]. Interestingly, no significant survival difference was observed between smokers and non-smokers, likely due to limited smoking history data in TCGA, such as pack-years, cessation duration, and smoking initiation age. Furthermore, treatment regimens, which were not considered, may also impact outcomes. Future studies

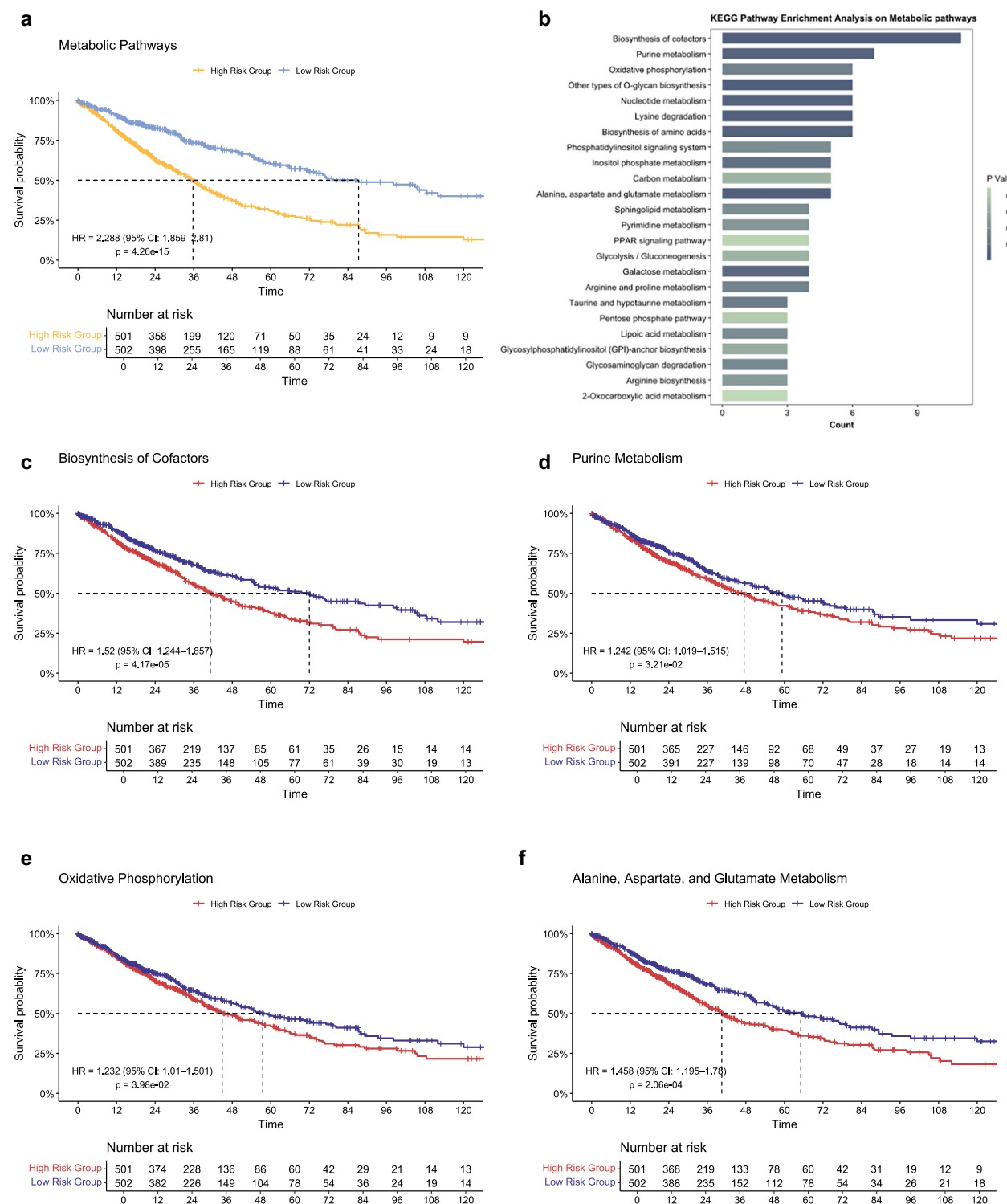

**Fig. 6 | Survival analysis of tRF-targeted metabolic pathways. a** Kaplan-Meier survival curve illustrating the prognostic significance of genes enriched in overall metabolic pathways. Dashed lines indicate the median survival threshold, corresponding to a 50% survival probability for the respective patient cohorts at specific time points. The tables below the plot display the number of patients at risk over the observation period. Statistical significance was determined using the log-rank test. HR hazard ratio, CI confidence interval. **b** KEGG pathway enrichment analysis identifying specific metabolic pathways associated with the enriched genes. Statistical significance is indicated by a p-value scale ranging from 0.05 (light green) to

0.01 (gray). **c–f** Kaplan-Meier survival curves comparing high-risk and low-risk groups based on pathway activity scores for key metabolic pathways, including biosynthesis of cofactors, purine metabolism, oxidative phosphorylation, and alanine, aspartate, and glutamate metabolism. Dashed lines represent the median survival threshold, corresponding to a 50% survival probability for the respective patient cohorts at specific time points. The tables below each plot show the number of patients at risk over the observation period. Statistical significance was determined using the log-rank test. HR hazard ratio, CI confidence interval.

should incorporate detailed smoking and treatment data, and use prospective, multi-omics approaches to improve tRNA-based survival models.

The correlation analysis suggests two signature tRNAs may regulate gene transcription through their derived tRNA fragments (tRFs). Supporting evidence from prior studies shows that variations in the expression of specific tRNA isodecoders may lead to changes in their corresponding tRFs within human tissues[44]. These changes are associated with pathological processes in infections, neurodegenerative diseases, and cancers, where increased tRNA turnover rates have been observed in tumor tissues compared to normal tissues, correlating tRF levels with cancer histological stages[45–47]. For example, 5' tRFs derived from *tRNA*$^{Glu}$, *tRNA*$^{Ser}$, *tRNA*$^{Leu}$, and *tRNA*$^{Arg}$ are notably overexpressed in prostate cancer cell lines, even exceeding the expression levels of many microRNAs (miRNAs)[48]. Similarly, in HeLa cells, 5' tRFs from *tRNA*$^{Lys}$, *tRNA*$^{Val}$, *tRNA*$^{Gln}$, and *tRNA*$^{Arg}$ are expressed at levels comparable to abundant miRNAs like miR-21 and the let-7 family[49]. Consistently, in our study, *5'-tRF-Val-CAC-2-1* and *5'-tRF-Val-CAC-1-5* demonstrated high regulation in NSCLC tissue and plasma samples, with a positive correlation to their originating tRNAs. Furthermore, *5'-tRF-Val-CAC* has been recognized as a potential biomarker in several cancers, including LUAD, oral squamous cell carcinoma, bladder cancer, etc[50–52]. Its role in cancer varies; for example, in gastric cancer, it critically regulates MAPK protein expression to influence tumorigenesis[53]; and in breast cancer, it participates in the tRF-17/THBS1/TGF-β1/Smad3 signaling pathway[54].

The KEGG results demonstrated the potential roles of tRF-regulated pathways in NSCLC progression and prognosis. Metabolic pathways identified in the KEGG enrichment analysis, including biosynthesis of cofactors, purine metabolism, oxidative phosphorylation, and alanine, aspartate, and glutamate metabolism, play fundamental roles in cellular energy production, biosynthetic processes, and maintenance of redox balance. These findings suggest that tRF-regulated pathways, particularly those involved in metabolism, contribute essentially to NSCLC progression and prognosis by supporting metabolic reprogramming, energy production, and tumor cell survival.

In previous lung cancer studies, the biosynthesis of cofactors is critical for activating enzymes involved in cancer metabolism, tumor proliferation, immune response, transcriptional regulation, and drug resistance[55,56]. For instance, *NFS1*, a gene enriched in this pathway, is vital in maintaining iron-sulfur clusters, which are essential for multiple cellular proteins. *NFS1* helps protect lung tumor cells from ferroptosis by mitigating oxidative damage, thereby promoting tumor survival[57]. Additionally, *NME3*, identified as a component of this pathway, enhances TLR5-mediated NFκB signaling in response to flagellin, thereby contributing to antitumor immunity in lung cancer patients[58]. Similarly, the purine metabolism pathway, crucial for nucleotide synthesis, is often upregulated in cancers to meet the increased demands for DNA and RNA synthesis during tumor proliferation[59]. This pathway has been explored as a therapeutic target in NSCLC, with inhibitors showing promise in reducing tumor growth. For example, *ATIC*, a gene enriched in this pathway, has been reported to promote LUAD cell growth and migration by enhancing Myc expression, underscoring its role in tumor progression[60]. Additionally, oxidative phosphorylation (OXPHOS), the primary ATP-generating pathway, is often reprogrammed in cancers, including NSCLC, to adapt to hypoxic tumor microenvironments. While many cancers predominantly rely on glycolysis[61], OXPHOS remains an essential mechanism for energy production, especially in NSCLC, supporting tumor cell division, migration, and invasion[62]. For instance, *ATP5F1B*, encoding a mitochondrial ATP synthase subunit, has been shown to inversely correlate with aerobic glycolysis rates in cancer cells. This highlights mitochondrial bioenergetic alterations as key contributors to glucose metabolism in NSCLC[63]. Furthermore, *UQCRC2*, another gene in this pathway, has been identified as a prognostic biomarker for LUAD in previous studies[64]. The alanine, aspartate, and glutamate metabolism pathway is integral to amino acid biosynthesis, nitrogen balance, and energy generation via the tricarboxylic acid (TCA) cycle[65,66]. Dysregulation of this pathway can disrupt nitrogen metabolism, driving the metabolic reprogramming commonly observed in NSCLC[67]. Genes such as *GLS2*, enriched

in this pathway, play a protective role by reducing oxidative stress in NSCLC cells[68]. Additionally, *ASNS* has been implicated in promoting lung tumor cell metastasis via the Wnt pathway and mitochondrial functions, even in the absence of endogenous asparagine production[69].

The diagnostic potential of tRNAs in NSCLC has been highlighted in the research; nevertheless, certain aspects still require improvement. First, while small RNA-seq can approximate tRNA expression levels, accurately quantifying mature tRNAs from our samples remains challenging due to their inherent secondary structures and extensive post-transcriptional modifications, which can compromise sequencing quality[34,70,71]. Additionally, the sequence similarity between precursor tRNAs, mature tRNAs, and tRNA fragments often results in ambiguous alignments during analysis[44,72]. Despite our efforts to fine-tune alignment and annotation parameters with stringent thresholds, ambiguities persist. The emerging technology of tRNA-seq offers a solution to these issues, potentially providing more accurate insights into tRNA biology as more data become available[73]. Furthermore, the development of refined tRNA mapping pipelines warrants comparison with our methods to enhance the precision of tRNA mapping[74]. Second, the small RNA-seq data available in public databases predominantly represent individuals of European descent, with limited representation from African Americans, Pacific Islanders, and Asians. This lack of diversity likely contributes to the unbalanced performance of our model and signature across racial groups, as our analysis showed consistently higher diagnostic accuracy in distinguishing between tumor and control samples in White and Asian populations, while the accuracy was relatively lower in the Black population. Addressing this limitation in future studies by expanding data collection to include a more diverse representation of racial groups will improve the generalizability of our findings and mitigate potential biases arising from genetic and racial differences. Third, our functional analyses indicate that the identified signature tRNAs may regulate gene expression post-transcriptionally through their derivatives, corroborating the prognostic value of specific tRF-targeted pathways, consistent with findings from other cancer studies. However, while our results highlight the potential significance of these pathways, the precise molecular mechanisms mediated by these signature tRNAs remain speculative. Future research should focus on elucidating these mechanisms through experimental validation and benchwork studies, which will not only enhance our understanding of their biological roles but also identify potential therapeutic targets for cancer treatment.

In a word, this study underscores the critical role of the tRNA signature in the advancement of diagnostic methodologies for NSCLC, providing encouraging paths for early, non-invasive detection methods.

## Data availability

Publicly available small RNA sequencing data and corresponding clinical information were obtained from the Gene Expression Omnibus (GEO) and The Cancer Genome Atlas (TCGA) databases. The GEO datasets include GSE110907, GSE62182, GSE83527, and GSE175462. TCGA data from lung cancer cohorts, specifically TCGA-LUAD and TCGA-LUSC, were accessed through the TCGA data portal. Direct links to these datasets are provided in the Methods section. The small RNA sequencing data from the RUSH cohort have been deposited in the database of Genotypes and Phenotypes (dbGaP) under accession number phs004166.v1.p1. The data will be made available to authorized researchers upon controlled access approval. Supplementary Data files accompanying this study include detailed metadata of all analyzed samples (Data S1), statistical filtering results for candidate tRNAs and tRFs (Data S2, S3), predicted target genes of selected tRFs (Data S4), and results of downstream functional enrichment analyses, including KEGG and GO pathway analyses of the predicted target genes (Data S5-7). These supplementary materials provide comprehensive support for the analyses and conclusions presented in the manuscript.

## Code availability

All statistical analyses in this study were performed using publicly available software packages, with optimized parameter settings detailed in the

Methods section. All analyses were conducted in accordance with the package's guidelines and its open-source licensing terms. The primary code is available at https://github.com/F-UH/Primary-Code/tree/master.

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

## Acknowledgements

This research was supported by grants from the National Institutes of Health (NIH): R01CA230514, R01CA223490, P20GM103466, P30CA071789, P20GM139753, U54GM138062, U54HG013243, T32DK137523, 1UE5HG013826, 3OT2OD032581-01S5-895 and 1OT2OD032581-02-PP90Y to Y.D. This work was also supported by NIH grants U24MD015970 and RCC-004UHI-Pilot, awarded to Y.F. Additional support was provided by NIH grant U54MD007601, awarded to M.N. We would like to express our appreciation to Orion S. Rivers and Tina Weatherby Carvalho of the University of Hawaii Pacific Biosciences Research Center Biological Electron Microscope Facility for their expert assistance with electron microscopy.

## Author contributions

Z.F.: Conceptualization, Methodology, Formal analysis, Visualization, Writing—Original Draft; M.N.: Validation, Investigation, Resources; G.D., A.A.A.-G., O.T.M.C., J.A.B.: Resources, Writing—Review & Editing; Z.G.: Methodology, Data Curation; H.Z., Y.C., T.G., G.L.: Data Curation; H.Y.: Project administration; L.W.*, Y.F.*, Y.D.*: Conceptualization, Supervision, Project administration, Funding acquisition.

## Competing interests

The authors declare no competing interests.

## Additional information

[1]Department of Quantitative Health Sciences, John A. Burns School of Medicine, University of Hawaii at Manoa, Honolulu, HI, USA. [2]Molecular Biosciences and Bioengineering Program, College of Tropical Agriculture and Human Resources, University of Hawaii at Manoa, Honolulu, HI, USA. [3]Genomics and Bioinformatics Shared Resource, University of Hawaii Cancer Center, Honolulu, HI, USA. [4]The Queen's Medical Center, Honolulu, HI, USA. [5]Pali Momi Medical Center, Honolulu, HI, USA. [6]Pathology Core Shared Resource, University of Hawaii Cancer Center, Honolulu, HI, USA. [7]RUSH University Medical Center, Chicago, IL, USA. [8]Illinois Institute of Technology, Chicago, IL, USA. [9]Pacific Center for Genome Research, University of Hawaii Cancer Center, Honolulu, HI, USA.
✉e-mail: lwu@cc.hawaii.edu; fuy@hawaii.edu; dengy@hawaii.edu

