## [Transparent Peer Review file · Communications Medicine]

Liquid Biopsy Diagnostics for Non-Small Cell Lung Cancer via Elucidation of tRNA Signatures

Corresponding Author: Professor Youping Deng

Version 0:

Reviewer comments:

Reviewer #1

(Remarks to the Author)

In the article with the title "Elucidation of Robust tRNA Signatures Through Multicenter Small RNA-Seq Data Analysis: Advancing Liquid Biopsy Diagnostics for Non-Small Cell Lung Cancer", the authors propose a six-tRNA signature for Non-Small Cell Lung Cancer (NSCLC) screening. Numerous statistical approaches are incorporated and functional regulations of selected tRNA fragments are proposed. Furthermore, the signature was identified in tissue samples and validated in exosomes deriving from patients and healthy individuals. The findings appear robust; yet some improvements are required:

Major comments:

1. In figure 2G in the RUSH group, tRNA-Lys-CTT-3-1 appears downregulated in contrast to all other groups. How is this observation evaluated by the authors?
2. A more in-depth explanation of the forwardSearch and backwardSearch algorithms appears essential.
3. It is advised to further present the survival results in the results section and not only in the discussion section.
4. I would recommend further highlighting the fact that the signature was identified in tissue samples and validated in exosomes.
5. I would propose to include the number of patients that are used in the figures. For example in the boxplots and ROC curves.
6. Please clarify the number of samples that were sequenced.
7. Please clarify the data that are incorporated in the T-score.

Minor comments:

8. The colored lines in the ROC curves could be thicker.
9. The common nomenclature of miRNAs is for example miR-30e-3p and not mir-30e-3p.
10. In table 2 I think a spelling error exists as the variable should be Smoking_history.

Reviewer #2

(Remarks to the Author)

The article delves into the analysis of small RNA sequencing data obtained from 1446 tissue samples in a public database to identify diagnostic tRNA signatures for Non-Small Cell Lung Cancer (NSCLC). Additionally, internal data from 233 plasma exosome samples from a prospective cohort were used to validate these findings. The evaluation of the diagnostic effect was conducted using the Area Under the Curve (AUC) index. Various clinical and demographic variables were scrutinized to assess characteristics, while the molecular functionality of tRNA features was further explored through survival analysis and functional studies.

Major Comments:

The timing of plasma sample collection lacks elucidation. While the article mentions the health or disease state of patients, it fails to detail whether patients had undergone anti-tumor treatments like immunotherapy, which can alter the lung cancer microenvironment.

The focus on tRNA and tRNA fragments (tRF) expression analysis aims to innovate a non-invasive liquid biopsy method for early NSCLC diagnosis. Nevertheless, the functional roles of these tRNA fragments and their mechanisms in gene regulation remain partially understood, necessitating further research for a comprehensive understanding of their

involvement in tumorigenesis.

Although the study underscores the diagnostic potential of tRNA characteristics in NSCLC, previous research, such as the findings from PMID: 36387119, has also spotlighted the diagnostic role of tRNA in NSCLC. This suggests the need for an in-depth exploration of exosomal tRFs as promising diagnostic biomarkers.

Incorporating data from databases like GSE110907, GSE62182, GSE83527, GSE175462, and TCGA is crucial. However, inconsistencies and missing information ("NA" entries) within these databases, and the large number of datasets used, may lead to errors during tRNA feature discovery and validation. Data collation and stringent exclusions are imperative to uphold result accuracy.

Lung tissue data fetched from public databases, including various sources like GSE datasets and TCGA, lack adequate clarification about the sequencing methods employed. Detailing the sequencing methods is essential to ensure data uniformity and accurate analysis, given the potential discrepancies arising from different sequencing techniques.

Minor Comments:

The validation cohort predominantly sourced from a specific geographic region or population may introduce selection bias, potentially impacting the findings' generalizability and applicability.

Sole reliance on the limma method of the R package for exploring tRF expression data limits methodological diversity. Multiple validation techniques should be employed to substantiate the findings.

Tables containing symbols for representation, particularly those with overlapping data denoting "1," could benefit from clearer illustrations to enhance result readability and comprehension.

Reviewer #3

(Remarks to the Author)
Summary

There is a need for a non-invasive diagnostic for NSCLC to facilitate early diagnosis and monitor response to treatment. tRNA expression profiles assayed from exosomes isolated from the blood have been shown to be associated with NSCLC diagnosis. The authors present a tRNA signature derived from blood exosomes, capable of discriminating NSCLC and healthy patients. They show that the expression profiles of three of the tRNAs correlate with prognosis. They also demonstrate that two of the signature tRNAs may have regulatory influence via tRFs.

Review

The authors present a novel tRNA signature derived from the application of a published meta-analysis approach to multiple small RNA expression dataset. They follow up with survival analysis validation and functional pathway inference via correlated tRF expression. The utilisation of multiple tumour/control LUAD and LUSC datasets including their own, is novel and adds significance to the derived signature. The follow up survival analysis demonstrates the signatures clinical relevance. This work will be of interest to the field.

I do raise the following concerns that should be addressed prior to publication.

Major comment

The authors chose to mix adenocarcinoma and squamous histologies. The underlying biology of these two pathologies is quite different and an analysis separating these two NSCLC types is warranted. Following on from this, the authors state a notable disparity and homogeneity in tRNA expression profiles between control and tumour groups (Figure 1B). The dispersion observed in the tumour samples across PC2 suggests there is variation in the tRNA profiles across samples. There is also overlap between a significant percentage of the control and tumour samples. I wonder whether the mixed picture of tRNA profile similarity within control and tumour sample groups is due to the two different histologies? A PCA separating the two histologies would be informative. A more detailed representation of the homogeneity in tRNA expression across the datasets and tissue sample batches is needed to meet the claims stated in the manuscript.

The authors show a good correlation between two signature tRNAs and two tRFs. They go on to identify pathways whose genes are potentially regulated by these two tRFs via an RNAhybrid analysis. The significance of these pathways in NSCLC progression and tRNA expression is over stated given the evidence. It would be interesting to see whether the expression of these pathways also correlate with survival in the same way as the tRNA signature. This would add weight to the involvement of these pathways in tumour associated tRNA expression.

Minor comments

The Authors apply a low gene count filter prior to signature construction and survival analysis. They exclude genes absent in

over 50% of the samples. This filter, as described, will remove genes expressed in up to 49% of the samples, thus potentially excluding important discriminatory genes. The application of this filter needs to be explained.

It is unclear from the figure legend which values are presented in the boxplots in Fig2B and D.

Key references are missing in the body of the text. For example there is no reference indicator for the MetaIntegrator tool used in the construction of the signature (line 135)

Version 1:

Reviewer comments:

Reviewer #1

(Remarks to the Author)

The Authors have well revised their manuscript, in line with the Reviewers' comments. The revised manuscript is much improved.

Reviewer #2

(Remarks to the Author)

The authors answered well to the comments. All our concerns have been addressed.

Reviewer #3

(Remarks to the Author)

Dear Reviewer #1,

We sincerely appreciate your detailed review of our manuscript. We have taken your feedback into careful consideration and have made the necessary revisions. For a comprehensive response to each of your comments, please see the attached document.

Major comments:

1. In figure 2G in the RUSH group, tRNA-Lys-CTT-3-1 appears downregulated in contrast to all other groups. How is this observation evaluated by the authors?

Response: We appreciate the reviewer's observation regarding the downregulation of tRNA-Lys-CTT-3-1 in the RUSH group. (We supplemented PRC curves in Figure 2 and now the heatmap has been shown in Figure 2J.) We acknowledge this difference and believe it may be due to sample heterogeneity within the RUSH cohort, potentially reflecting inherent biological variability among individual samples. Another possible explanation lies in the difference in sample types—while other cohorts are primarily based on tissue samples, the RUSH group comprises plasma-derived exosomal samples. Previous studies have demonstrated that the gene expression profiles of exosomal RNA can slightly differ from those of tissue RNA due to differences in their biogenesis, processing, and functional roles in intercellular communication. These factors may account for the observed differences in tRNA-Lys-CTT-3-1 expression in the RUSH group.

Additional samples, particularly with consistent sample types across cohorts, would be beneficial to confirm and evaluate this observation, enabling more comprehensive validation in future research. Nevertheless, we emphasize that the overall pattern and trend of T-scores for the entire set of signatures remain consistent across all groups, underscoring the robustness of our findings and supporting the general validity of our conclusions on tRNA expression trends.

2. A more in-depth explanation of the forwardSearch and backwardSearch algorithms appears essential.

Response: Thank you for your insightful comment. We apologize for any confusion regarding the algorithmic steps involved in selecting the final 6 tRNAs. Here's a detailed explanation of the forwardSearch and backwardSearch algorithms:

The forwardSearch and backwardSearch algorithms follow greedy algorithm-based, iterative approaches designed to optimize the selection of tRNAs with the highest discriminatory power, especially in distinguishing cases from controls in the datasets used.

Starting with a single candidate tRNA, the forwardSearch algorithm gradually adds more tRNAs from the original pool (117 tRNAs were found to be differently expressed using the MetaIntegrator). At each step, forwardSearch evaluates which tRNA, when added, maximizes

the weighted AUC across datasets. The weighted AUC represents the sum of the AUC for each dataset, adjusted by the number of samples, thereby focusing on maximizing the performance across the largest number of samples rather than on any specific dataset. The forwardSearch function stops adding tRNAs when it reaches a preset threshold in units of weighted AUC. We set the threshold as 0 by default to ensure the optimization process will continue iteratively adding genes to the set until no further increase in the weighted AUC is observed.

Conversely, the backwardSearch algorithm starts with the full set of candidate tRNAs and iteratively removes one tRNA at a time, removing those whose exclusion leads to the greatest improvement in weighted AUC. Like forwardSearch, backwardSearch is also based on a greedy algorithm, removing only the tRNAs that contribute the least to the model's predictive performance. The stopping threshold for backwardSearch is also based on weighted AUC, ensuring that the final selection remains optimized for performance across datasets.

The intersection of both algorithms' results, a commonly used technique in gene set optimization, produced the final six-tRNA signature for our model. This intersection ensures a robust selection of tRNAs that consistently maximize weighted AUC while reducing potential redundancy.

Given the non-convex nature of gene selection optimization, the forwardSearch and backwardSearch algorithms provide locally optimized solutions rather than a global optimum. These algorithms rely on a step-by-step selection or elimination of genes based on their specific contribution to the AUC, resulting in a set of signature that is fine-tuned to maximize predictive accuracy across multiple datasets. This method helps ensure that the selected tRNAs provide maximum discriminatory power without overwhelming the model with excess features.

We hope this explanation clarifies the methodology, and we have revised the manuscript to better explain the *Material and Method* (pages 19-20, lines 566-576).

3. It is advised to further present the survival results in the results section and not only in the discussion section.

Response: Thank you for this valuable suggestion. In response, we have expanded the *Result* section to include detailed survival analysis findings, such as the identification of three prognostic tRNAs, the calculation of the cumulative risk score, and multivariate Cox regression validation (pages 8-9, lines 213-230). Previously, these details were included only in the Discussion.

We also revised the *Discussion* to focus on interpreting these findings, comparing them to existing literature, and addressing study limitations (page 13, lines 362-377). This reorganization clarifies the presentation of results and strengthens the interpretive discussion.

4. I would recommend further highlighting the fact that the signature was identified in tissue samples and validated in exosomes.

Response: We appreciate the reviewer's insightful suggestion. In response, we have added a dedicated paragraph in the discussion section that highlights the identification of the six-tRNA signature in tissue samples and its successful validation in exosomes (page 12, lines 339-350). This addition underscores the robustness and potential clinical applicability of the signature as a versatile biomarker across different sample types.

5. I would propose to include the number of patients that are used in the figures.

Response: Thank you for your valuable suggestion. We have updated the figures to include the number of patients used in the box plots. The ROC and PRC curves share the same sample size as the corresponding box plots within the same phase or analysis of the same factor. These revisions ensure consistency and enhance the interpretability of the data (Fig. 2, 3).

6. Please clarify the number of samples that were sequenced.

Response: We apologize for any confusion regarding the sample count. All 233 samples collected from RUSH University were sequenced with small RNA sequencing as part of this research. We have revised the manuscript to clarify the number of sequenced samples in the *Material and Method* section (Page 17, lines 499-507).

7. Please clarify the data that are incorporated in the T-score.

Response: Thank you for highlighting this point. To clarify, the T-score was calculated using the expression levels (TPM normalized) of the tRNA signature genes across all samples in the specific dataset. Specifically, the calculation involves three steps:

1. Geometric Mean Calculation:

The geometric means of up-regulated (S_{up}) and down-regulated (S_{down}) tRNAs for each sample i were calculated as:

$$GeoMean_{up}^i = \left(\prod_{j \in S_{up}} TPM_j^i \right)^{\frac{1}{|S_{up}|}}$$
$$GeoMean_{down}^i = \left(\prod_{j \in S_{down}} TPM_j^i \right)^{\frac{1}{|S_{down}|}}$$

2. Preliminary Score:

The preliminary score was computed as:

$$Score^i = GeoMean_{up}^i - GeoMean_{down}^i$$

3. Standardization:

$$T-score^i = \frac{Score^i - \mu}{\sigma}$$

where μ and σ are the mean and standard deviation of all preliminary scores.

We have revised the *Material and Method* section to clarify the data included in the T-score calculation (page 20, lines 584-598).

Minor comments:

8. The colored lines in the ROC curves could be thicker.

Response: Thank you for your careful suggestion. We have refined the curves of ROC and PRC to be more thicker and clearer (Fig. 2, 3).

9. The common nomenclature of miRNAs is for example miR-30e-3p and not mir-30e-3p.

Response: Thank you for pointing this out. We have revised the manuscript to ensure that all miRNA nomenclature follows the standard format, using "miR" instead of "mir" (e.g., miR-30e-3p).

10. In table 2 I think exists as the variable should be Smoking_history.

Response: Sorry for the mistake. We have corrected the spelling of the variable to "Smoking_history" in Table 2 to ensure the accuracy of Table 2.

Dear Reviewer #2,

We are grateful for your detailed review of our manuscript. We have thoroughly addressed each of your comments and have made corresponding revisions to our manuscript. Please refer to the attached file for more detailed response.

Major Comments:

1. The timing of plasma sample collection lacks elucidation. While the article mentions the health or disease state of patients, it fails to detail whether patients had undergone anti-tumor treatments like immunotherapy, which can alter the lung cancer microenvironment.

Response: Thank you for highlighting this important consideration. The plasma samples collected from RUSH University were obtained between February 2010 and June 2019, with all blood samples collected at the time of initial diagnosis. While some patients underwent surgical biopsy during the diagnostic phase, other relevant treatments—including surgery, radiation, or chemotherapy—were conducted only after blood sampling. Though data on immunotherapy is not available for these patients, the timing of blood collection at diagnosis suggests that any potential impact of anti-tumor treatment on the blood samples should be minimal. We have clarified these details in the manuscript to address the timing and context of sample collection (pages 16-17, lines 479-487).

2. The focus on tRNA and tRNA fragments (tRF) expression analysis aims to innovate a non-invasive liquid biopsy method for early NSCLC diagnosis. Nevertheless, the functional roles of these tRNA fragments and their mechanisms in gene regulation remain partially understood, necessitating further research for a comprehensive understanding of their involvement in tumorigenesis.

Response: Thank you for your insightful comment. To address this, our study explores the regulatory potential of tRFs through a comprehensive analysis of their target pathways, particularly focusing on their prognostic implications in NSCLC.

We conducted survival analyses on the expression levels of specific metabolic pathways targeted by tRFs, demonstrating the significant prognostic value of these pathways in NSCLC progression (Fig. 6, S5). Kaplan-Meier analysis revealed that low-risk scores, derived from the activity of tRF-targeted gene-enriched pathways, consistently correlated with longer survival spans, emphasizing their potential role in tumor development and prognosis (Fig. 6A-F). These pathways, identified via KEGG enrichment analysis, include biosynthesis of cofactors, purine metabolism, oxidative phosphorylation, and alanine, aspartate, and glutamate metabolism (Fig. 6B). Together, these pathways support the metabolic demands of rapidly proliferating tumor cells by enabling cellular energy production, biosynthetic processes, and redox balance.

Further insights into the functional significance and mechanisms of these pathways, including detailed functions of genes of specific pathway roles in NSCLC, are supplemented in the *Discussion* section of the manuscript (pages 14-15, lines 395-431). These findings enhance our understanding of the complex regulatory functions of tRFs in NSCLC and underscore their

potential as biomarkers and therapeutic targets. While these results provide a solid foundation, further *in vitro* and *in vivo* studies will be essential to comprehensively elucidate the mechanisms by which tRFs regulate these pathways and contribute to tumorigenesis.

3. Although the study underscores the diagnostic potential of tRNA characteristics in NSCLC, previous research, such as the findings from PMID: 36387119, has also spotlighted the diagnostic role of tRNA in NSCLC. This suggests the need for an in-depth exploration of exosomal tRFs as promising diagnostic biomarkers.

Response: Thank you for highlighting the relevance of prior studies, such as PMID: 36387119, which focused on the diagnostic role of tRNA fragments (tRFs) in NSCLC. While we acknowledge the importance of exploring the underlying mechanisms—an aspect we plan to address in future research—we believe our study offers significant advancements over previous work.

First, unlike the cited study that specifically targets tRFs, our research investigates tRNAs as diagnostic signatures and includes tRFs as part of a broader analysis of tRNA function. Additionally, the cited study's sample size was relatively small, involving 495 samples with RNA-seq findings from only 10 samples. In contrast, our study utilizes a much larger dataset of 1,679 samples, with signature identification conducted on a discovery cohort of over 1,000 samples, enhancing the robustness and statistical power of our results.

Feature selection in the previous study was performed manually, whereas we applied machine-learning methods to ensure a systematic and data-driven approach, leading to more reliable feature selection. In the aspect of signature performance, the cited study reported a ROC of 0.74 for its tRF biomarker, while our model achieved an improved ROC of 0.8, reflecting better diagnostic performance.

Furthermore, the prior study primarily distinguished cancer samples from healthy controls without examining comparisons between cancerous and benign samples—a more clinically relevant distinction that our research includes, adding value to potential diagnostic applications. Finally, while the previous study focused solely on diagnostic value, our work also explores the prognostic implications of the tRNA signature.

These distinctions underscore the comprehensive nature and advancements of our study. We have highlighted these unique contributions throughout the manuscript to clearly convey the improvements and broader implications of our research compared to prior studies.

4. Incorporating data from databases like GSE110907, GSE62182, GSE83527, GSE175462, and TCGA is crucial. However, inconsistencies and missing information ("NA" entries)

within these databases, and the large number of datasets used, may lead to errors during tRNA feature discovery and validation. Data collation and stringent exclusions are imperative to uphold result accuracy.

Response: Thank you for underscoring the importance of data collation and careful exclusions to uphold result accuracy. To develop a robust model, we integrated data from multiple studies to enhance the comprehensiveness of our analysis. We recognize that inconsistencies and missing data in public databases pose inherent challenges. To address these, we employed the limma package to minimize batch effects across datasets by modeling batch effects as linear additive covariates, preserving meaningful biological signals. Additionally, the Leave-One-Out (LOO) algorithm was utilized during feature selection to prevent any single dataset from disproportionately influencing the results. After identifying the tRNA signature, we validated its performance across diverse demographic and clinical subgroups, including age, sex, cancer subtype, disease stage, and smoking history, to ensure its generalizability (Fig. 3).

During our review of dataset details, we supplemented missing information where possible. For example, we identified that all samples in dataset GSE110907 were derived from Asian patients (Table 1). This additional racial information enabled us to assess the performance of the tRNA signature across different racial groups, adding depth to our evaluation (Fig. S6).

Our analysis revealed high diagnostic accuracy for the tRNA signature in both White and Asian populations, with AUC values exceeding 0.84. However, the performance was relatively lower for Black individuals, likely due to smaller sample sizes and underrepresentation in the dataset. This discrepancy underscores the need for larger and more diverse cohorts in future studies to ensure equitable model performance across all racial groups. These findings have been incorporated into the revised manuscript in the *Results* and *Discussion* sections to provide a balanced and transparent evaluation of the signature's applicability (page 8, lines 201-206; page 15, line 443-451).

5. Lung tissue data fetched from public databases, including various sources like GSE datasets and TCGA, lack adequate clarification about the sequencing methods employed. Detailing the sequencing methods is essential to ensure data uniformity and accurate analysis, given the potential discrepancies arising from different sequencing techniques.

Response:

I appreciate you pointing this out. To clarify, data retrieved from public databases, including GEO datasets of GSE110907, GSE62182, GSE83527, GSE175462 and TCGA dataset, were all conducted with miRNA-seq based on whole transcriptome sequencing on the Illumina HiSeq

2000 platform (*Homo sapiens*). This information and detailed references have been added to the manuscript to ensure transparency and consistency in data analysis (page 17, lines 502-507).

The small RNA sequencing of 233 plasma-derived RNA samples was conducted using the Illumina NextSeq 500 system. Although the HiSeq 2000 and NextSeq 500 platforms differ slightly in technology and throughput, both platforms are Illumina-based systems that ensure comparability by generating similar read lengths, data quality, and sensitivity in miRNA detection. Both sequencing methods follow standard protocols for miRNA library preparation and quality control, such as adapter trimming, enrichment of small RNAs, and alignment to reference genomes.

Minor Comments:

6. The validation cohort predominantly sourced from a specific geographic region or population may introduce selection bias, potentially impacting the findings' generalizability and applicability.

Response: Thank you for your insightful feedback. Thank you for your insightful feedback. We recognize that sourcing the validation cohort predominantly from specific geographic regions or populations could introduce selection bias, potentially affecting the generalizability of our findings. To address this concern, we assessed the signature's performance across different racial groups, observing strong diagnostic accuracy in White and Asian populations (AUC values >0.84). However, the performance was limited in the African American population (AUC of 0.5), with no significant distinction between cancer and control groups.

This discrepancy likely results from unbalanced sample sizes across racial groups, with a predominance of White participants in the dataset. Additionally, sourcing primarily from specific geographic regions may influence the representativeness of the validation cohort. These findings underscore the importance of expanding data collection to include more diverse and balanced populations in future studies to enhance the robustness and applicability of the signature. This analysis and its implications have been incorporated into the *Results* and *Discussion* sections of the manuscript (page 8, lines 201–206; page 15, lines 443–451).

7. Sole reliance on the limma method of the R package for exploring tRF expression data limits methodological diversity. Multiple validation techniques should be employed to substantiate the findings.

Response: Thank you for your comment, and I apologize for any confusion. To clarify, the limma package was used solely for batch effect correction across datasets to ensure data consistency in our integrated analysis. Differential expression analysis and signature identification of tRNAs were conducted using the MetaIntegrator package, which combines

effect sizes across studies to identify robust biomarkers. As described in the *Methods and Materials* section, we further refined the signature using the Leave-One-Out (LOO) algorithm along with forwardSearch and backwardSearch functions.

To validate the identified signature, we used complementary approaches, including independent validation on an external cohort, demographic and clinical subgroup analyses, and various statistical tests (e.g., Mann-Whitney U, ROC or PRC analyses, and multivariate Cox regression) to assess the diagnostic and prognostic potential of the tRNA signature. These comprehensive validation steps enhance the validity and generalizability of our findings.

8. Tables containing symbols for representation, particularly those with overlapping data denoting "1," could benefit from clearer illustrations to enhance result readability and comprehension.

Response: Thank you for the valuable suggestion. We have revised the table to improve clarity. Specifically, we changed the adjusted HR for the reference factor from "1.00(1.00-1.00)" to "Reference" to more clearly indicate its role as the baseline in the analysis. Additionally, we have clarified this in the table legend to enhance interpretability (Table 2). We have also addressed factors with no significant association in the *Discussion* section of the manuscript (page 13, lines 362-377).

Dear Reviewer #3,

We sincerely appreciate your comprehensive review of our manuscript and your insightful feedback. We have taken each of your comments into careful consideration and have revised the manuscript accordingly. Please refer to the attached file for more detailed response.

Major comment

1. The authors chose to mix adenocarcinoma and squamous histologies. The underlying biology of these two pathologies is quite different and an analysis separating these two NSCLC types is warranted. Following on from this, the authors state a notable disparity and homogeneity in tRNA expression profiles between control and tumour groups (Figure 1B). The dispersion observed in the tumour samples across PC2 suggests there is variation in the tRNA profiles across samples. There is also overlap between a significant percentage of the control and tumour samples. I wonder whether the mixed picture of tRNA profile similarity within control and tumour sample groups is due to the two different histologies? A PCA separating the two histologies would be informative. A more detailed representation of the homogeneity in tRNA expression across the datasets and tissue sample batches is needed to meet the claims stated in the manuscript.

Response: Thank you for your valuable comment. We conducted a PCA analysis on the LUAD, LUSC, and control groups. The results revealed a significant difference in tRNA expression profiles between cancer samples (LUAD and LUSC combined) and the control group. However, no significant difference was observed between LUAD and LUSC subtypes (Figure S8).

This finding suggests that the histological subtypes within NSCLC—primarily LUAD and LUSC—share highly similar tRNA expression profiles, even though tumor development is associated with distinct tRNA dysregulation compared to normal controls. This similarity is likely due to shared oncogenic pathways and cellular processes that both subtypes depend on for tumor growth and survival, leading to overlapping molecular signatures. Furthermore, tRNAs, as fundamental components of protein synthesis and cellular metabolism, are often dysregulated across cancer types but may exhibit limited variation between closely related subtypes within the same organ. This observation is further supported by the tRNA analysis in webserver of DBtrend (<https://trend.pmrc.re.kr/>), which also showed no significant difference in tRNA expression between LUAD and LUSC cohorts.

These findings confirm the feasibility of studying tRNA as diagnostic biomarkers across the NSCLC spectrum, encompassing both LUAD and LUSC. We have revised the manuscript to reflect this clarification on the implications of these findings (page 5, lines 130-133; page 11, lines 308-316).

2. The authors show a good correlation between two signature tRNAs and two tRFs. They go on to identify pathways whose genes are potentially regulated by these two tRFs via an RNAhybrid analysis. The significance of these pathways in NSCLC progression and tRNA expression is overstated given the evidence. It would be interesting to see whether the expression of these pathways also correlate with survival in the same way as the tRNA signature. This would add weight to the involvement of these pathways in tumour associated tRNA expression.

Response: Thank you for the inspiring suggestion. To address your comment, we conducted a comprehensive survival analysis to evaluate whether the expression of metabolic pathways potentially regulated by the two signature tRNAs and their derived tRFs correlates with survival outcomes in NSCLC patients. Utilizing RNA expression data from TCGA-LUAD and TCGA-LUSC, we analyzed 92 genes enriched in metabolic pathways identified through RNAhybrid and KEGG enrichment analysis (Fig. 6, S8).

The survival analysis of these 92 genes, taken collectively, demonstrated significant prognostic value, as shown in Figure 6A. KEGG enrichment analysis further highlighted that these tRF-binding genes are involved in critical metabolic pathways, including biosynthesis of cofactors, purine metabolism, oxidative phosphorylation, and alanine, aspartate, and glutamate metabolism (Figure 6B). To refine our findings, we conducted survival assessments on specific pathways.

Pathways such as biosynthesis of cofactors, purine metabolism, oxidative phosphorylation, and alanine, aspartate, and glutamate metabolism exhibited significant prognostic differences, with lower pathway activity scores consistently associated with longer survival spans (Figures 6C-F).

These results align with previous studies that suggest metabolic pathways play a fundamental role in NSCLC progression and prognosis by supporting metabolic reprogramming, energy production, and tumor cell survival. For example, the biosynthesis of cofactors is critical for activating enzymes involved in cancer metabolism and immune responses, while purine metabolism and oxidative phosphorylation provide essential nucleotides and energy for rapid tumor growth. Similarly, the alanine, aspartate, and glutamate metabolism pathway contribute to amino acid biosynthesis and nitrogen balance, supporting tumor development.

While these analyses provide strong evidence of the involvement of tRF-regulated pathways in NSCLC progression, further *in vitro* and *in vivo* studies are needed to elucidate the precise molecular mechanisms by which tRFs regulate these pathways and contribute to tumorigenesis. These findings have been incorporated into the manuscript, with detailed discussions on the specific pathways supplemented in the *Results* and *Discussion* section (page 10, lines 267-277; pages 14-15, lines 395-431).

Minor comments

3. The Authors apply a low gene count filter prior to signature construction and survival analysis. They exclude genes absent in over 50% of the samples. This filter, as described, will remove genes expressed in up to 49% of the samples, thus potentially excluding important discriminatory genes. The application of this filter needs to be explained.

Response: Thank you for your insightful observation. The low gene count filter, set at 50%, was applied to enhance the robustness and reliability of the downstream analyses. Specifically, we excluded genes absent in more than 50% of the samples to minimize noise and focus on features with consistent and detectable expression levels across the dataset. This threshold is balanced between retaining informative features and removing sparsely expressed genes, which are less likely to contribute robust and reproducible discriminatory power, especially in clinical screening applications aimed at the general population.

This approach aligns with established practices in sequencing data analysis, as the 50% threshold is a widely adopted filtering criterion for bulk RNA-seq data. It reduces the possibility of false connections caused by low-expression genes while guaranteeing that downstream studies focus on genes with consistent expression patterns. This ultimately enhances the generalizability and applicability of the identified signature across diverse patient populations.

Furthermore, the large size and diversity of our dataset ensure that important genes contributing to classification and survival predictions are retained, even with the application of this filter. We

have updated the manuscript to clarify the reason for this filtering step and its impact on the analysis (page 18, lines 529-531).

4. It is unclear from the figure legend which values are presented in the boxplots in Fig2B and D.

Response: Thank you for pointing out the ambiguity in the legend. We have refined the legend for the boxplots in the updated Fig. 2 to provide greater clarity. The boxplots represent the T-scores across different datasets, comparing tumor and control cohorts during the discovery, hold-out validation, and independent validation phases, respectively. Statistical comparisons were conducted using the Mann-Whitney U test, with significance levels denoted as follows: “*” for $p < 0.05$, “**” for $p < 0.01$, and “***” for $p < 0.001$.

5. Key references are missing in the body of the text. For example there is no reference indicator for the MetaIntegrator tool used in the construction of the signature.

Response: Thank you for bringing this to our attention. We have thoroughly reviewed the manuscript and supplemented the missing key references, including the citation for the MetaIntegrator tool used in the construction of the signature. The relevant reference has been added as the 66th citation (page 19, lines 553–556) to ensure proper attribution and enhance the manuscript's completeness and credibility.